# Is time a variable like the others in multivariate statistical downscaling and bias correction?

Robin Yoann[1,*] and Vrac Mathieu[2,*]

[1]Météo-France, 42 avenue Gaspard-Coriolis, 31057, Toulouse, France
[2]Laboratoire des Sciences du Climat et de l'Environnement (LSCE-IPSL), CEA/CNRS/UVSQ, Université Paris-Saclay,
Centre d'Etudes de Saclay, Orme des Merisiers, 91191 Gif-sur-Yvette, France
[*]The two authors contributed equally to this study

**Correspondence:** Robin Y. (yoann.robin@meteo.fr), Vrac M. (mathieu.vrac@lsce.ipsl.fr)

**Abstract.** Bias correction and statistical downscaling are now regularly applied to climate simulations to make then more usable for impact models and studies. Over the last few years, various methods were developed to account for multivariate – inter-site or inter-variable – properties in addition to more usual univariate ones. Among such methods, temporal properties are either neglected or specifically accounted for, i.e., differently from the other properties. In this study, we propose a new multivariate approach called "Time Shifted Multivariate Bias Correction" (TSMBC), which targets to correct the temporal dependency in addition to the other marginal and multivariate aspects. TSMBC relies on considering the initial variables at various times (i.e., lags) as additional variables to correct. Hence, temporal dependencies (e.g., auto-correlations) to correct are viewed as inter-variable dependencies to be adjusted and an existing multivariate bias correction (MBC) method can then be used to answer this need. This approach is first applied and evaluated on synthetic data from a Vector Auto Regressive (VAR) process. In a second evaluation, we work in a "perfect model" context where a Regional Climate Model (RCM) plays the role of the (pseudo-) observations, and where its forcing Global Climate Model (GCM) is the model to be downscaled/bias corrected. For both evaluations, the results show a large reduction of the biases in the temporal properties, while inter-variable and spatial dependence structures are still correctly adjusted. However, increasing too much the number of lags to consider does not necessarily improve the temporal properties and a too strong increase in the number of dimensions of the dataset to correct can even imply some potential instability in the adjusted/downscaled results, calling for a reasoned use of this approach for large datasets.

## 1  Introduction

Climate and Earth System models (ESM) and their simulations are the main physical tools to investigate the potential future evolutions of the climate system (e.g., Flato et al., 2013; Kirtman et al., 2013). They are clearly indispensable to test how different scenarios of greenhouse gas emission trajectories might induce climate changes and, thus, to try to anticipate potential impacts of those changes (e.g., Shukla et al., 2019). Although such elaborated models contain many relevant and complex processes characterizing the climate properties and their dependencies, the numerical simulations they generate are often tainted with biases and disagreements with respect to observations. Those can stem: (i) from the spatial resolution of the

simulations – from 200km×200km for global climate models, GCMs, down to a few km for regional climate modes, RCMs

–, usually too low compared to needs of impact models that may require very local or high-resolution input climate data, e.g., kilometre, hundreds of meters or below, down to the weather stations (e.g., Chen et al., 2011; Maraun and Widmann, 2018); and/or (ii) from inherent biases in the model simulations, due to parameterizations, or processes not or poorly represented (e.g., McFarlane, 2011). Even with respect to observations or reanalysis data at the models' spatial resolution, the simulations can present biases, i.e., disagreements on some statistical properties (e.g., mean, variability, distribution) that differ from reference

data (e.g., Christensen et al., 2008).

Those issues make that impact models (such as for hydrology, energy, environment, etc.) cannot directly employ the climate simulations as input (e.g., Teutschbein and Seibert, 2012; Chen et al., 2013). Indeed, most impact models are calibrated on observational data and the use of biased climate data as input could result in inappropriate or even false impact projections.

To overcome those spatial or inherent biases, various statistical post-processing methods have been developed for downscaling and/or bias correction of the climate simulations. The common idea is to statistically transform the numerical simulations – over a historical period – in such a way that some properties become equal or close to those of a chosen reference dataset (e.g., Gudmundsson et al., 2012). The statistical downscaling or correction transformation is then usually supposed valid under other climate conditions and applied to changed (e.g., future) climate. The obtained downscaled/bias corrected climate data can then

serve as input into impact models (e.g., Teutschbein and Seibert, 2012; Galmarini et al., 2019; Bartók et al., 2019; Chen et al., 2021, among many others).

Over the last two decades, many such post-processing methods were developed, either in a "Perfect prognosis" (PP) context, generally for downscaling (DS), or in a "Model Output Statistics" (MOS) one, generally for bias correction (BC) – see e.g.,

Vaittinada Ayar et al. (2015) or Maraun and Widmann (2018) for the differences between the two, or Vrac and Vaittinada Ayar (2017) for a combination of PP and MOS. Note that, in practice, BC methods (i.e., MOS) are often used to perform downscaling (see, e.g., Thrasher et al., 2012; Hempel et al., 2013; Frieler et al., 2017). Hence, in the following, we will simply refer to "bias correction" (hereafter BC) even for downscaling purposes. Up to recently, most of the BC methods were designed to work on "univariate" data, representing one climate variable at one given location. In such a univariate context, PP approaches

include, e.g., linear regressions (Jeong et al., 2012), or non-linear ones such as polynomial regressions and artificial neural network (Xiaoli et al., 2008), or stochastic weather generators (Wilks, 2012). In a similar univariate context, MOS methods can vary from very simple methods correcting only the mean via anomalies adjustments (e.g., Xu, 1999, with the "anomaly" method), or the variance (e.g., Eden et al., 2012; Schmidli et al., 2006, with the "simple scaling" method) to more complete and widespread methods such as the "quantile-mapping" approach that corrects the whole univariate distribution and therefore all

statistical moments (including mean and variance) of the variable of interest. Quantile-mapping (e.g., Haddad and Rosenfeld, 1997; Déqué, 2007) is certainly the most used 1d-BC method by practitioners and numerous variants have been developed, e.g., by Vrac et al. (2012) to account for non-stationarity, by Kallache et al. (2011) for modelling of extremes, by Cannon et al. (2015) to ensure preservation of the climate change signal, or by Vrac et al. (2016) specifically for precipitation including

occurrences, among many other variants.

However, the univariate correction of simulations (i.e., one variable at a time and one site at a time) may not be enough. Indeed, the use of several 1d-corrections separately for different physical variables and/or sites will not correct the dependencies between them (Vrac, 2018). The consequence is that, if the simulations are biased in their spatial and/or inter-variable correlations (or more generally in their dependencies), most of the 1d-BC methods will conserve those biases (Vrac, 2018;

François et al., 2020). If an impact (e.g., hydrological) model needs realistic dependencies between its input climate variables, the use of univariate BC may not provide corrections realistic enough (e.g. Boé et al., 2007). More generally in climate sciences, the accurate modelling of dependencies is a key aspect for proper assessments and projections of compound events and their associated risks (e.g. Leonard et al., 2014; Zscheischler et al., 2018; Bevacqua et al., 2019). This characteristics strongly motivated the development of BC methods accounting for multivariate links between the variables and/or the sites of interest.

As detailed in Vrac (2018) or François et al. (2020), Multivariate BC methods can be categorized in three approaches: the "marginal/dependence" correction approach – separating the correction of the marginal distributions and that of the dependence structure (e.g., Bárdossy and Pegram, 2012; Cannon, 2018; Vrac, 2018) –, the "successive conditional" correction one – applying successive corrections conditionally on the previously corrected variables (e.g., Piani and Haerter, 2012; Dekens et al., 2017) – and the "all-in-one" correction approach – correcting altogether the univariate marginal properties and their

dependence structures (Robin et al., 2019). Those methodologies have been applied to correct either inter-variable dependence only (i.e., multiple variables but spatial dependence not accounted for), spatial dependence only (one variable at multiple locations) or even both (see François et al., 2020, for a review and comparison of the methods).

One main conclusion provided in the MBC comparison study brought by François et al. (2020) is that if inter-site and inter-

variable properties can be reasonably adjusted by multivariate bias correction (MBC) methods, the temporal properties are usually not taken into account in most MBC procedures. As any MBC will necessarily modify the rank chronology of the simulations to perform the multivariate correction (Vrac, 2018), it results that temporal properties and dependencies issued from the MBC output are transformed with respect to the raw simulations but are not necessarily closer to those of the reference data, depending on the method itself, its setting and the variables of interest (François et al., 2020). Therefore, accounting

for temporal properties (i.e., autocorrelations or even cross-autocorrelations) when performing MBC is needed to increase the realism of the multivariate corrected time series. This is a major concern for impact studies: for example, hydrological models are very sensitive to the atmospheric forcing used as inputs and particularly, among others, to biases in the chronology of events (e.g., Raimonet et al., 2017; Bhuiyan et al., 2018). Some works tried to correct temporality properties in addition to (spatial or inter-variable) MBC. For example: Johnson and Sharma (2012) used a nesting 1d-BC model across various timescales;

this was extended to include inter-site structures by Mehrotra and Sharma (2015) and then by Mehrotra and Sharma (2016) to account also for inter-variable dependence. In those studies, time dependence was specifically modeled via (multivariate) autoregressive models with periodic parameters. Another recent study by Vrac and Thao (2020) proposed an MBC method for both inter-site and inter-variable properties, where the temporal aspects are accounted for through an analog-based method

applied to multivariate sequences of ranks.

All those approaches, although different, share the fact that they try to correct temporal properties by a (parametric or non-parametric) model that is specific to the variable "time". In other words, they separate time from the other variables (variables at various locations) of interest. However, one can wonder if there is a real need for such a specificity. Indeed, let us take an example, in one dimension for sake of clarity (this can be easily generalised to $n$ dimensions). Say we have a time series of

a climate variable $\boldsymbol{X}$ to be corrected (e.g., temperature) at one location, for $t = 1, ..., N : \boldsymbol{X}_{1:N} = (x_1, ..., x_N)$. If we apply a 1d-BC method to the sample $\boldsymbol{X}_{1:N}$, most of the BC methods will preserve the rank correlation of $\boldsymbol{X}$ (Vrac, 2018). Now, instead of applying a 1d-BC, if we duplicate and lag – say of one time step – the time series $\boldsymbol{X}_{1:N}$, we obtain a shifted time series noted $\boldsymbol{X}^S_{1:N-1} = \boldsymbol{X}_{2:N} = (x_2, ..., x_N)$. If a multivariate bias correction method is able to correct the dependence between two univariate time series, say $\boldsymbol{X}$ and $\boldsymbol{Y}$, when applied to the bivariate time series

$$(\boldsymbol{X}, \boldsymbol{X}^S)_{1:N-1} = \begin{pmatrix} x_1 & x_2 \\ x_2 & x_3 \\ \vdots & \vdots \\ x_{N-1} & x_N \end{pmatrix},$$   (1)

it should then be able to correct the dependence between the univariate time series $\boldsymbol{X}$ and $\boldsymbol{X}^S$, i.e., it should be able to correct the lag-1 temporal dependence of $\boldsymbol{X}$. As far as we know, this specific approach of considering time as any other statistical variable in an MBC procedure has never been investigated. This is therefore the goal of the present study. More specifically, the aim is to investigate (i) if this "time shifted multivariate bias correction" (TSMBC) approach does correct temporal properties

(e.g., auto-correlations); (ii) if and how it impacts other properties (e.g., inter-site and inter-variable); (iii) if this TSMBC is able to work for more than lag-1.

To do so, the rest of the manuscript is organized as follows: Section 2 describes the climate and synthetic data used in this study and provides detailed explanations about the proposed TSMBC approach. Section 3 investigates the method based on

artificially simulated multivariate dataset obtained from a Vector Autoregressive model (VAR). Then, TSMBC is applied to correct daily temperature and precipitation simulated from a climate model and the results are given in Section 4. Finally, conclusions are reminded and perspectives are discussed in Section 5.

## 2  Data and methodology

### 2.1  Data: Climate simulations and VAR processes

To apply any (M)BC method, it is necessary to dispose of a dataset of reference – i.e., supposed to be as close as possible to real observed climate – and a dataset of simulations (e.g., stemming from a GCM) that are biased with respect to the reference

dataset.

Here, the climate simulations to be corrected are daily temperature and precipitation times series over South-East of France,
for the time period 1951-2010, extracted at a $1.25^o \times 2.5^o$ spatial resolution from the IPSL-CM5A-MR global climate model
(Marti et al., 2010; Dufresne et al., 2013) developed at "Institut Pierre Simon Laplace" (IPSL). The simulations are forced by
historical conditions up to 2005 and then by RCP8.5 scenario conditions from 2006. The region of interest is presented in Fig 1
and corresponds to a zone where correlations between precipitation and temperature as well as autocorrelations are difficult
to model due to various geographical constrains, such as Mediterranean influences and 3 mountain ranges (the Pyrenees, the
Massif Central, the Alps), which can generate high precipitation events known as "Cevennol events" (see, e.g. Delrieu et al.,
2005).

Regarding the reference dataset, regional climate simulations are used in this study, instead of observational or reanalysis data.
Those are EURO-CORDEX daily temperature and precipitation from the KNMI-RACMO22E regional climate model (KNMI,
2017) developed at "The Royal Netherlands Meteorological Institude" (KNMI) and forced by the larger-scale IPSL simula-
tions. The same time period and region as for the IPSL simulations were extracted but with a $0.11^o \times 0.11^o$ spatial resolution.
Note that the extracted region of interest is small in comparison to the initial EURO-CORDEX domain (Jacob et al., 2014)
over which the RCM simulations were performed. This kind of "perfect model experiment", i.e., considering simulations as
"pseudo-observations", is now a common approach to assess a downscaling / bias correction methods, (see, e.g., Charles et al.,
2004; Vrac et al., 2007; Frost et al., 2011; Bürger et al., 2012; Grouillet et al., 2016).

The GCM data are then interpolated with a nearest neighbors method to the $0.11^o \times 0.11^o$ spatial resolution of the RCM to
apply the TSMBC approaches. The (GCM and reference) data over the period 1951-1980 will serve to calibrate the proposed
TSMBC method, while projections and evaluations will be performed over 1981-2010. Moreover, the calibration/projection
steps are realized on a seasonal basis, i.e., for the 4 seasons separately to account for specific seasonal features of the biases.
For sake of clarity, only the summer (JJA) results are shown in the manuscript, winter (DJF) results being provided as supple-
mentary materials.

In addition to the evaluations that will be done based on those climate simulations, a preliminary analysis will first be performed
on synthetic data, i.e., data artificially generated from statistical models. Here, a "Vector autoregressive" (VAR) process is
employed. A VAR process is a multivariate AutoRegressive (AR) process (i.e., allowing multivariate data) modelling the
statistical link between the components of a vector (i.e., multivariate data) when they change in time. In the following, a
VAR is used to generate multivariate time series $(\boldsymbol{X}_t)_{t=1,...,N}$ where each $\boldsymbol{X}_t$ is a vector of dimension $d$ (e.g., $d = 2$ with

temperature and precipitation at one location, or with temperature only at two locations), with prescribed autocorrelations up to a certain lag $s$, such that:

$$155 \quad \boldsymbol{X}_n = \boldsymbol{b} + \sum_{l=1}^{s} \mathbf{A}_l \boldsymbol{X}_{n-l} + \boldsymbol{\epsilon} \tag{2}$$

where $s$ is the order of the VAR process for which the autocorrelation is accounted for, $\boldsymbol{b}$ is the $d$-dimensional intercept vector, $\mathbf{A}_l$ are matrices of coefficients and $\boldsymbol{\epsilon}$ is a $d$-dimensional noise following a multivariate Gaussian law with $0$ mean vector and covariance matrix $\boldsymbol{\Sigma}$. Two VAR processes are generated: one used as the large-scale or biased simulations and the other one used as the reference.

To generate such synthetic data and analyse them in a comprehensive way, the dimension $d$ has been chosen equal to 2 and lag $s$ equal to 3. The sampling is performed based on Equation (2): the first $s$ vectors (i.e., from time 1 to $s$) are initialised and the VAR process allows generating new values for the $d$ components of the vector at time $s+1$. Also, in order to be in a realistic case, i.e., close to our climate simulations context, the parameters $\boldsymbol{b}$, $(\mathbf{A}_l)_{l=1\ldots3}$ and $\boldsymbol{\Sigma}$ have been estimated from a

165 VAR fitted to temperature at two opposite grid points in summer (from the GCM or RCM, for VARs representing biased or reference data respectively) in the region of interest. The parameters obtained for these two VARs are then used to simulate 2500 synthetic data for each of the two VAR processes, approximately corresponding to the number of summer days over the available calibration period. The univariate and bivariate probability density functions of those simulations are plotted in Fig. 2(a), in red for VAR data to be corrected and in blue for the reference VAR.

## 2.2   The TSMBC approach: autocorrelations seen as correlations

The main philosophy of the proposed "multivariate time shifted bias correction" (TSMBC) approach has been briefly introduced (with lag 1) in Eq. (1) of the introduction section. We now described the approach in a more general way and with more details. Say we dispose of a $d$-dimensional time series (i.e., matrix) $\mathbf{X}_{1:N} = (\boldsymbol{X}^1, \boldsymbol{X}^2, ..., \boldsymbol{X}^d)_{1:N}$, where each $\boldsymbol{X}^i_{1:N}$ is a

175 $N$-dimensional vector (i.e., time series of length $N$): $(x^i_1, ..., x^i_N)^T$, where the superscript $T$ on a vector denotes its transpose. The idea proposed and tested in this article consists in applying a multivariate bias correction onto the matrix $\mathbf{M}(s)$ constituted by the gathering of the initial $\mathbf{X}_{1:N}$ time series and those obtained after shifting it by different lags up to lag $s$:

$$\mathbf{M}_{\mathbf{X}}(s) = \left( (\boldsymbol{X}^1, \boldsymbol{X}^2, ..., \boldsymbol{X}^d)_{1:N-s}, (\boldsymbol{X}^1, \boldsymbol{X}^2, ..., \boldsymbol{X}^d)_{2:N-s+1}, ..., (\boldsymbol{X}^1, \boldsymbol{X}^2, ..., \boldsymbol{X}^d)_{s+1:N} \right) \tag{3}$$

or equivalently:

$$
\mathbf{M_X}(s) = \left( \begin{pmatrix} x_1^1 & x_1^2 & \cdots & x_1^d \\ x_2^1 & x_2^2 & \cdots & x_2^d \\ \vdots & \vdots & \ddots & \vdots \\ x_{N-s}^1 & x_{N-s}^2 & \cdots & x_{N-s}^d \end{pmatrix}, \begin{pmatrix} x_2^1 & x_2^2 & \cdots & x_2^d \\ x_3^1 & x_3^2 & \cdots & x_3^d \\ \vdots & \vdots & \ddots & \vdots \\ x_{N-s+1}^1 & x_{N-s+1}^2 & \cdots & x_{N-s+1}^d \end{pmatrix}, \ldots, \begin{pmatrix} x_{s+1}^1 & x_{s+1}^2 & \cdots & x_{s+1}^d \\ x_{s+2}^1 & x_{s+2}^2 & \cdots & x_{s+2}^d \\ \vdots & \vdots & \ddots & \vdots \\ x_N^1 & x_N^2 & \cdots & x_N^d \end{pmatrix} \right),
$$

(4)

which can then be grouped by variable (dimension) and thus reordered as:

$$
\mathbf{M_X}(s) = \left( \begin{pmatrix} x_1^1 & x_2^1 & \cdots & x_{s+1}^1 \\ x_2^1 & x_3^1 & \cdots & x_{s+2}^1 \\ \vdots & \vdots & \ddots & \vdots \\ x_{N-s}^1 & x_{N-s+1}^1 & \cdots & x_N^1 \end{pmatrix}, \begin{pmatrix} x_1^2 & x_2^2 & \cdots & x_{s+1}^2 \\ x_2^2 & x_3^2 & \cdots & x_{s+2}^2 \\ \vdots & \vdots & \ddots & \vdots \\ x_{N-s}^2 & x_{N-s+1}^2 & \cdots & x_N^2 \end{pmatrix}, \ldots, \begin{pmatrix} x_1^d & x_2^d & \cdots & x_{s+1}^d \\ x_2^d & x_3^d & \cdots & x_{s+2}^d \\ \vdots & \vdots & \ddots & \vdots \\ x_{N-s}^d & x_{N-s+1}^d & \cdots & x_N^d \end{pmatrix} \right).
$$

(5)

Such a transformation can be made to create $\mathbf{M_Y}(s)$ and $\mathbf{M_X}(s)$, the lagged matrices obtained respectively from the reference dataset and the biased one to correct. An MBC method can thus be applied to correct the multivariate properties of $\mathbf{M_X}(s)$ with respect to $\mathbf{M_Y}(s)$ as reference. The result of the MBC is therefore a matrix $\mathbf{M_Z}(s)$ of the same size as $\mathbf{M_X}(s)$. The question to answer now is: "From this result MBC matrix, how to extract/reconstruct a $N \times d$ matrix $\mathbf{Z}$ that corresponds to proper multivariate (time included) corrected data?". There are indeed various possibilities for that and they may not be equivalent.

In this study, we propose a method based on a reconstruction *by rows*. For illustration, let us take this example of a bivariate dataset $\mathbf{X}$ and its associated matrix $\mathbf{M_X}(1)$, i.e., including a lag-1 shift:

$$
\mathbf{X} = \begin{pmatrix} 1 & 10 \\ 2 & 20 \\ 3 & 30 \\ 4 & 40 \\ 5 & 50 \\ 6 & 60 \end{pmatrix}, \mathbf{M_X}(1) = \begin{pmatrix} \mathbf{1} & \mathbf{10} & \mathbf{2} & \mathbf{20} \\ 2 & 20 & 3 & 30 \\ \mathbf{3} & \mathbf{30} & \mathbf{4} & \mathbf{40} \\ 4 & 40 & 5 & 50 \\ \mathbf{5} & \mathbf{50} & \mathbf{6} & \mathbf{60} \end{pmatrix}
$$

In this example, the bold rows of the matrix $\mathbf{M_X}(1)$ (its rows 1, 3 and 5) can be used to reconstruct the original dataset $\mathbf{X}$. More generally, a row $s$ of $\mathbf{M_X}(1)$ is a vector corresponding to a portion of $\mathbf{X}$, which continues at row $s+2$, and which continues at row $2s+3$, etc. Concatenating these rows, we can re-construct the dataset $\mathbf{X}$. Applying this method to the matrix $\mathbf{M_Z}(s)$, we can reconstruct a corrected dataset $\mathbf{Z}$. Because the same operation can be applied by starting at the second row (and any row

between 1 and $s+1$), the reconstruction depends of the choice of the starting row. However, even if the values are repeated in the lagged matrix, no repeated values can appear in the final reconstruction (i.e., multivariate corrections). It is also worth noting the initial mapping (i.e.,going from $\mathbf{X}$ to $\mathbf{M_X}$ is injective, and that, in general, an inverse mapping is not. However, with the suggested reconstruction (i.e., inverse mapping), when a starting row is chosen, the inverse mapping gives a unique time series and is thus injective. Moreover, the choice of a starting row $r > 1$ omits the first $r - 1$ values in the final reconstruction. This leads us to wonder about the influence of the choice of the starting row. These points will be investigated with the VAR processes. Our approach is summarized in the Alg. 1.

Note that a reconstruction "*by column*" could also be performed: each column of $\mathbf{M_X}(1)$ being a sub-column of $\mathbf{X}$, it could then be used to reconstruct the original dataset. However, preliminary analyses showed that this approach does not allow to correct auto-correlations, as the temporal dependence structure in each row is not corrected and mostly corresponds to that of the model to adjust (not shown). Hence, only the TSMBC approach "by rows" is investigated in the following.

---

**Algorithm 1** Summary of the TSMBC method "by rows"

---

**Require:** A $N_\mathbf{X} \times d$ biased dataset $\mathbf{X}$, where $N_\mathbf{X}$ is the number of time steps for $\mathbf{X}$, and $d$ the number of variables

**Require:** A $N_\mathbf{Y} \times d$ reference dataset $\mathbf{Y}$, where $N_\mathbf{Y}$ is the number of time steps for $\mathbf{Y}$

**Require:** A multivariate bias correction method, dOTC in this paper

**Require:** A lag $s$ and a starting row $r$

Build the $(N_\mathbf{X} - s) \times d(s+1)$ matrix $\mathbf{M_X}(s)$ from $\mathbf{X}$ with Eq. (5)

Build the $(N_\mathbf{Y} - s) \times d(s+1)$ matrix $\mathbf{M_Y}(s)$ from $\mathbf{Y}$ with Eq. (5)

Apply the multivariate bias correction method between $\mathbf{M_X}(s)$ and $\mathbf{M_Y}(s)$, generating the $(N_\mathbf{X} - s) \times d(s+1)$ matrix $\mathbf{M_Z}(s)$

Apply the reconstruction with the row method with starting row $r$ to the matrix $\mathbf{M_Z}(s)$. This generates the correction $\mathbf{Z}$.

**return** The $N_\mathbf{X} \times d$ multivariate correction $\mathbf{Z}$.

---

Finally, because TSMBC uses an underlying MBC, potentially any MBC method can be used, as MBCn (Cannon, 2018) or R2D2 (Vrac and Thao, 2020). Here, the MBC method used is the "dynamical Optimal Transport Correction" (dOTC) developed by Robin et al. (2019). While most of the bias correction methods build a mapping between the biased and the reference dataset, dOTC infers a probability distribution $\mathbb{P}$ such that $\mathbb{P}(\boldsymbol{x}, \boldsymbol{y})$ is *the probability that $\boldsymbol{y}$ is the correction of $\boldsymbol{x}$*. This probability distribution is inferred – with optimal transport methods, see Appendix A and Villani (2008); Santambrogio (2015) – between the biased and reference dataset in calibration period (representing the bias), and between the biased dataset in calibration and projection periods (representing the evolution). The "evolution" distribution is then transferred along the "bias" distribution to construct a correction in the reference world with an evolution similar to that of the biased data. Note that this method seeks to preserve the evolution of the model while reducing bias. A brief reminder about dOTC is given in appendix A, while all details can be found in Robin et al. (2019).

## 3 Results: Synthetic VAR data

In this section we test our TSMBC method on synthetic data, generated from two VAR processes (see sub-section 2.1). A biased dataset and a reference one – noted respectively $\mathbf{X}$ and $\mathbf{Y}$ – are drawn from two VAR processes fitted from two time series of temperature of the GCM/RCM. TSMBC is applied to correct $\mathbf{X}$ with respect to $\mathbf{Y}$, i.e., to generate a corrected dataset $\mathbf{Z}$.

Because the reconstruction step preserves the dependence structure, we propose to test which part of the correction is due to the underlying method (here dOTC), and which part is due to the reconstruction. To do so, a second underlying bias correction method is then used as a benchmark. It corresponds to a very naive method: the correction is randomly drawn from the reference dataset, i.e., for any $\boldsymbol{x} \in \mathbf{X}$, the correction is given by a random value $\boldsymbol{y}$ generated according to the distribution of $\mathbf{Y}$. In practice, values from $\mathbf{Y}$ are resampled. Note that this naive method corresponds to dOTC with a transport plan given by the product probability distribution between the biased/reference dataset, see App. A. We note this multivariate method a "Random Bias Correction" (RBC), and TSMBC is thus available in two versions for this evaluation: TSMBC(dOTC) and TSMBC(RBC). For these two methods, we test in a first part the influence of the choice of the starting row, and in a second part the influence of the choice of the lag.

### 3.1 Influence of the choice of the starting row

We fix the number of lags $s = 10$ and we compute the correction for each of the two methods with a starting row $r \in \{1, 3, 6, 9, 12\}$. Note that, since here $s = 10$, TSMC applied with $r = 12$ will provide the same results as for $r = 1$ but will ignore the first row. The corrections are noted $\mathbf{Z}_r$.

To measure the similarity of the corrections from different starting rows, we compute the matrix of Pearson correlations between the pair $\mathbf{Z}_{r_1}/\mathbf{Z}_{r_2}$. We also compute the correlations with $\mathbf{X}$ and $\mathbf{Y}$ to compare the corrections with the biased and reference datasets. The results are presented in Fig. 3a for TSMBC(dOTC) and in Fig. 3b for TSMBC(RBC). Only the first dimension is represented, the second dimension – which gives similar results – and the associated $p$-values are given in Fig. S1 of the supplementary material. We can see for TSMBC(dOTC) that all corrections are highly correlated between them – with values close to 1 – whereas for TSMBC(RBC) no significant correlation appears. This indicates that, whatever the chosen starting row for TSMBC, the results are very close to each other. Remark that for TSMBC(dOTC) the corrections stay highly correlated with $\mathbf{X}$. It is an effect of the dOTC method, which tries to preserve as much as possible the temporal properties of the model simulations to be corrected (Robin et al., 2019; François et al., 2020). On the other hand, no correlation appears with $\mathbf{Y}$. This was expected. Indeed, there is no reason for the bias corrected data to be correlated to the reference. If the BC procedure is efficient, corrected and reference time series can be seen as generated based on the same statistical distributions and/or properties but independently. Hence, they are not correlated.

From this experiment we can conclude first that the choice of the starting row has only a very marginal influence on the correction. Therefore, from now on, we use the integer part of $r = \frac{s+1}{2}$ as starting row. Secondly, the TSMBC(RBC) method

produces many different corrections, extremely weakly correlated between them. Even if, by construction, the TSMBC(RBC) is able to provide satisfactory inter-variable dependence structures and correlations, it is not able to provide proper temporal statistics, highlighting the importance of the choice of the underlying bias correction method.

## 3.2 Influence of the choice of the lag

In this section the starting row is fixed at $r = \frac{s+1}{2}$, and we vary the number of lags $s \in \{3, 5, 10, 20, 100\}$. This generates the corrections $\mathbf{Z}_s$ for the two methods TSMBC(dOTC) and TSMBC(RBC).

As for the previous sub-section, the correlations between corrections, reference and biased dataset are computed, and repre-
sented in Fig. 3c,d for the first variable, the second variable and $p$-values are given in Fig. S2. We can see for TSMBC(dOTC) that all corrections are highly correlated – and correlated with the biased dataset –, with a correlations decreasing with $s$. This shows that the corrections are similar, even when increasing the lag to a reasonable extent. As for the previous section, the method TSMBC(RBC) provides many different corrections, different from each other and thus difficult to link together. Furthermore, we have added in Fig. 2a the density of the corrections with TSMBC(dOTC) and with TSMBC(RBC) in green
and black respectively. The two methods adjust correctly the density and the dependence structure of the reference dataset. Therefore the differences seen in Fig. 3c,d come only from their capability to adjust the temporal structure.

The (cross) auto-correlations between the 2 dimensions for various lags are also given in Fig. 2b-e. On each panel, we can see that the results for the lag 0 – which corresponds to classic correlations – are closed to the references for all corrections, which is confirmed by the Fig. 2a. For non-zero lags, the (cross) auto-correlations of corrections with TSMBC(dOTC) are closed to
the reference one, validating the method. For TSMBC(RBC), the (cross) auto-correlations become close to the reference if $s$ is large enough. This shows that the ability to correct the (cross) auto correlation when $s$ is small comes from dOTC, and not only from the reconstruction part. When s increases, TSMBC(RBC) tends to adjust the temporal properties correctly but this is due to the reconstruction that replaces the biased dataset by the reference one for large $s$.

Generally, from the synthetic VAR dataset, we can see the ability of the TSMBC approach to correct the (cross) auto-correlations. The choice of the starting row has little influence on the final corrections, and we fix it now at $\frac{s+1}{2}$. We also highlighted the importance of dOTC as underlying bias correction method (compared to the naive random approach), as the ability to properly correct the temporal structure does not come only from the reconstruction step of TSMBC.

## 4   Results: DS/BC of temperature and precipitation

We now apply the TSMBC method with the underlying dOTC method to the bias correction/downscaling of the IPSL GCM simulations with respect to the RCM simulations taken as references. Following the strategy proposed by François et al. (2020), four kinds of corrections are applied and analysed:

- Each variable and grid point are corrected independently. This approach will be referred to as "L1V" (Local 1 Variable). The BC method employed here is dOTC in its univariate version (when $s = 0$);

- The dependence between temperature and precipitation (i.e., inter-variable dependence) is taken into account in the correction, but not the spatial dependence. This approach is noted "L2V" (Local 2 Variables), and employes the bivariate version of dOTC (when $s = 0$);

- The spatial dependence is corrected, but not the relations between temperature and precipitation. This approach is noted "S1V" (Spatial 1 Variable) and uses dOTC in a 16 (longitudes) $\times$ 13 (latitudes) $= 208$ dimensional configuration (when $s = 0$);

- All dependencies (i.e., inter-variable and spatial) are corrected. This approach is noted "S2V" (Spatial 2 Variables), and dOTC has thus a 2 (variables) $\times$ 16 (longitudes) $\times$ 13 (latitudes) $= 416$-dimensional configuration (when $s = 0$).

Furthermore, for each of these approaches, we apply TSMBC to account for various lags, up to some maximum lags: 0-day (i.e., corresponding to dOTC, without any lag), 5-day and 10-day lags, noted dOTC, TSMBC-5 and TSMBC-10, respectively. Hence, we have finally 12 correction approaches, with dimensions varying from 1 (dOTC without any kind of dependence) to $2 \times 16 \times 13 \times (10 + 1) = 4576$ (MSTBC-10 correcting spatial and inter-variable dependencies and temporal dependence up to lag 10-day). A summary of the dimension of each method is given in Tab. 1.

Recall that only the results for summer are given in the rest of this article (winter results are provided as supplementary materials), the calibration period is 1951-1980, and the validation/projection period is 1981-2010.

## 4.1 Bias reduction in marginal properties: mean, standard deviation, auto-correlation

We start by controlling the ability of the different methods to reduce the bias of the first two statistical moments: the mean (noted $\mathbb{E}$) and the standard deviation (noted $\sigma$). Noting $\mathbf{Z}$ any correction, and $\kappa$ the statistics of interest – such as the mean $\mathbb{E}$, the standard deviation $\sigma$, or the lag-$s$ autocorrelation $\rho_s$ –, we compute the following criterion $BR_\kappa$ to characterize the bias reduction:

$$BR_\kappa = 1 - \left| \frac{\kappa(\mathbf{Z}) - \kappa(\text{RCM})}{\kappa(\text{GCM}) - \kappa(\text{RCM})} \right|. \tag{6}$$

This criterion lives in the interval $[-\infty, 1]$. A value of 1 indicates a perfect correction. Note that if the raw simulations already have a DCP set close to the reference, its Wasserstein distance will be near zero. If the correction gives also a Wasserstein distance very close to zero, then the relative reduction of bias $BR_\kappa$ can have a very strong negative value if $\kappa(\mathbf{Z}) > \kappa(\mathbf{GCM})$, even if the absolute difference (i.e., $\kappa(\mathbf{Z}) - \kappa(\mathbf{GCM})$) is potentially very small. Boxplots of mean bias reduction ($BR_\mathbb{E}$) and

standard deviation bias reduction ($BR_\sigma$) criteria from all grid points are presented in Fig. 4 for each variable and correction. For the calibration period (first column) we can see a bias reduction (both in means and standard deviations) between 0.95 and 1 for the temperature, and between 0.8 and 1 for precipitation. The bias reduction slightly decreases when the dimension increases, indicating a "curse of dimensionality" problem. For the projection period, the bias reduction in mean temperature stay reasonable, between 0.7 and 1. However, for the precipitation the results are more contrasted. For 75% of the grid points, the reduction of the mean precipitation bias lies between 0.2 and 0.95 and that of the standard deviation bias between 0.6 and 1. Interestingly, the use of a TSMBC approach relying on dOTC implies bias reductions in means and standard deviations equivalent to those provided by the dOTC method, i.e., without "time shifted" consideration. In other words, TSMBC does not degrade the basic marginal properties of the corrections from dOTC. Note that the ability of the methods to reduce the biases can be strongly affected by the evolution of the GCM variables between the calibration and validation period. Indeed, this evolution is different from that of the RCM (i.e., here, reference) variables. As dOTC preserves the evolution of the GCM to be corrected, the resulting corrections for the projection period have necessarily properties different from the reference over the same period.

The same boxplot are now represented in Fig 5 for the lag-2 auto-correlation bias reduction $BR_{\rho_2}$. The couples tas/tas, pr/pr, tas/pr and pr/tas are, respectively, the correlations between temperature and lagged (i.e. past) temperature, precipitation and lagged precipitation, temperature and lagged precipitation, and precipitation and lagged temperature. Over the calibration period, the use of TSMBC clearly improves the (cross-) auto-correlation compared to dOTC. In the projection period, the TSMBC correction is better than the dOTC correction for the pr/pr couple. No clear improvement appears for the couples pr/tas and tas/pr, and for the couple tas/tas a degradation appears when dimension increases, related to the problems already described for the mean and standard deviation.

Figure 6 presents the maps of auto-correlations of precipitation at lag 1 (first two lines) and at lag 4 (last two lines). The calibration period corresponds to the first and third lines, and the projection period to the second and last lines. We focus here only on the L2V approach, but the results are equivalent for the others (not shown, except for temperature with L2V given in Fig. S4). Regarding the calibration period, it is clear that dOTC does not reproduce the RCM auto-correlation maps. In addition, the obtained auto-correlation maps are quite different from those of the GCM, justifying the need for TSMBC. Based on TSMBC-5 and TSMBC-10, the corrections are closer to the reference RCM. Using 10 lags instead of 5 does not show a clear improvement, and can sometimes even degrade the corrections. For the projection period, the situation is quite different and requires to compare the GCM, RCM and corrections, as well as their evolution between the calibration and projection periods. For the GCM, the lag-1 auto-correlation decreases between the two periods for the northern part, but increases in the southern part. As dOTC mostly reproduces the evolution of the model (see Robin et al., 2019), the same feature appears in the corrections over the projection period. TSMBC-5 or TSMBC-10 using dOTC as underlying MBC method, the same conclusion holds. This result is also true for the lag 4, with auto-correlations more noisy for TSMBC-5 and TSMBC-10 due to a higher

number of dimensions to consider.

Globally, TSMBC is able to reduce biases in means and standard deviations as well as dOTC but clearly improves the corrections of the auto-correlations. We now propose to further study the dependence structure of the corrections brought by TSMBC.

## 4.2 Bias reduction in dependencies: the $\mathcal{W}$-cross-auto-correlogram metric

The present sub-section targets the evaluation of the TSMBC corrections in terms of spatial structure of auto-correlations between variables and grid points. This requires a new tool: the $\mathcal{W}$-cross-auto-correlogram metric, based on the classic correlogram.

In order to evaluate spatial dependencies present in a univariate sample, correlograms (i.e., correlations expressed as function of the distance) are classically used (see, e.g. Vrac, 2018; François et al., 2020). To estimate it, the correlation between each pair of locations (or grid cells in the present study) is computed. Then the set of distance/correlation pairs (DCP) is divided into classes (e.g., 0-10km, 10-20km, etc) and the conditional mean correlation is calculated for each class. However, following an equivalent procedure, the calculation of correlation in a univariate context can be replaced by the auto-correlation for any lags, or, in a multivariate setting, by cross-auto-correlations between variables (e.g., correlation between temperature at time $t$ for one location and precipitation at time $t+\text{lag}$ for another location). This gives an auto-correlogram or a cross-auto-correlogram, respectively.

Figure 7 presents the scatter-plots of the DCP values, where correlations are cross-auto-correlations between temperature and precipitations for all pairs of grid points with lag 1 for the GCM data (panel a), the RCM (panel b), the corrections with dOTC (S2V, panel c) and with TSMBC-5 (S2V, panel d). In each panel, the black line is the estimated mean cross-auto-correlogram. It is clear that the cross-auto-correlograms – as mean conditional correlations given distances – are not quite representative of the DCP scatter-plot structures. Indeed, the DCP structures show large spreads that cannot be visible on simple lines. Moreover, those structures are different from one dataset to another, for example between the GCM data to be corrected and from the reference RCM data, while their associated cross-auto-correlogram (i.e., mean black lines) can appear relatively close to each other. However, the DCP scatter-plots of the dOTC (c) and TSMBC-5 (d) corrections exhibit a large improvement compared to the uncorrected GCM, with TSMBC-5 that seems better than dOTC, with more realistic spread and shape of the DCP set. To quantify this improvement, for the GCM, the RCM and all the corrections, all DCP sets are first computed for lags between 0 and 9, for auto-correlations (temperature/temperature and precipitation/precipitation) and for cross-auto-correlations (temperature/precipitation and precipitation/temperature). Then, the Wasserstein distance $\mathcal{W}$ (see App. B) is calculated and used as a metric to measure the difference between the DCP sets from the RCM (i.e., here, the reference) and the DCP sets of the GCM or the corrections. The Wasserstein distance (e.g., Santambrogio, 2015; Robin et al., 2019) is a distance between two multivariate distributions and can therefore be considered as an alternative to the Energy distance (Rizzo and Székely,

2016; François et al., 2020). Noting respectively $\mathrm{DCP_{RCM}}$, $\mathrm{DCP_{GCM}}$ and $\mathrm{DCP_Z}$ these DCP sets, we propose the following indicator, $BR_{\mathcal{W}}$, based on the Wasserstein distance, to measure the bias reduction in dependence with respect to the raw GCM:

$$BR_{\mathcal{W}} = 1 - \left| \frac{\mathcal{W}(\mathrm{DCP_Z}, \mathrm{DCP_{RCM}})}{\mathcal{W}(\mathrm{DCP_{GCM}}, \mathrm{DCP_{RCM}})} \right| \tag{7}$$

where $\mathcal{W}(\mathrm{DCP_Z}, \mathrm{DCP_{RCM}})$ is the Wasserstein distance between the DCP set from the corrections $\mathbf{Z}$ and that from the reference RCM data; and $\mathcal{W}(\mathrm{DCP_{GCM}}, \mathrm{DCP_{RCM}})$ the equivalent between the GCM and RCM DCP sets. A value of $BR_{\mathcal{W}}$ close

to 1 indicates that the chosen dependencies (correlation, auto-correlation or cross-auto-correlation) of the correction is close to the dependence structure of the reference RCM. We call the set of this indicator a *W-cross-auto correlogram*.

As the Wasserstein metric is sensitive to the scale of the multivariate data (here, the DCP sets) it is applied to, two normalizations of the DCP sets are proposed before the computation of the $BR_{\mathcal{W}}$ values. First, we normalize the correlation values

of the DCP sets between $-1$ and $1$, independently for each method and lag. This allows us to compare only the pattern of the DPC sets, by removing mean and scale biases. The results are shown in Fig. 8. Alternatively, a second type of normalization is performed, with a normalization common to all methods for a given lag but different for different lags. Hence, this normalization conditional to the lag allows us to compare the different bias correction methods and include both the intensities of the correlations and the DCP patterns while first normalization only accounts for the pattern of the DCP sets. The results are

presented in Fig. 9, where only the $BR_{\mathcal{W}}$ values coming from the same "column" (i.e., lag) can be compared.

Starting with the first normalization (for each method/lag separately), allowing to compare only the pattern of the DPC sets, in Fig. 8, we can first see that $BR_{\mathcal{W}}$ matrices for calibration and projection periods are relatively similar, indicating some robustness of the TSMBC method over projections. Generally speaking, for the local configurations (L1V and L2V), TSMBC

(5 or 10) is better than dOTC that does not account for temporal properties. This is true for almost all lags >0 and any $BR_{\mathcal{W}}$ matrix (tas/tas, tas/pr, pr/tas, pr/pr). However, for the spatial configurations (S1V and S2V), TSMBC does not seem to provide better results than dOTC, except for the tas/tas matrix where TSMBC strongly improves dOTC. Moreover, although dOTC provides similar results for the L1V and L2V configurations (potentially of poor quality for various lags), the use of S1V and S2V approaches within dOTC – i.e., without specifically accounting for temporal dependence – strongly improves the $BR_{\mathcal{W}}$

results. This indicates that imposing to account for spatial properties within dOTC can improve the correction of the auto-correlation and cross-auto-correlation patterns. Moreover, regarding the tas/tas and pr/pr $BR_{\mathcal{W}}$ matrices, the spatial versions (i.e., S1V and S2V) of the method appear largely better than the local (i.e., L1V and L2V) ones, especially for tas/tas for most of the lags. This was somehow expected since the S1V and S2V configurations also correct spatial dependencies. Regarding the $BR_{\mathcal{W}}$ matrices for tas/pr and pr/tas, results are better for L2V and S2V configurations than with L1V and S1V ones: ac-

counting for correlations and cross-auto-correlations between temperature and precipitation allows improving the results over configurations based on temperature and precipitation considered separately. Furthermore, it is not clear that increasing the number of lags in TSMBC significantly improves the results. Here, TSMBC-10 and TSMBC-5 provide quite similar $BR_{\mathcal{W}}$

values whatever the couple of variables of interest, with some non-systematic variations along the lags. However, globally, the DCP patterns are clearly improved by the TSMBC corrections, as shown by the positive $BR_{\mathcal{W}}$ values in Fig. 8.

Continuing with the $BR_{\mathcal{W}}$ results obtained based on the second type of normalization in Fig. 9, the results are more contrasted, as negative values are now visible. This traduces the fact that, although DCP patterns are globally improved by the various configurations of dOTC and TSMBC (as seen previously in Fig. 8), biases are present in the intensities of correlations and, when accounted for, degrades the $BR_{\mathcal{W}}$ values. However, as for the first normalization, the spatial configurations (S1V or

S2V) seem to reduce biases more than the local (L1V or L2V) configurations; the L2V and S2V approaches are in general better than the L1V and S1V ones for the cross-correlations matrices (i.e., involving both tas and pr): and the S1V versions appear better than the S2V ones for matrices with single variables (i.e., tas/tas and pr/pr). This latter remark is due to the fact that with the S2V approach, the complexity of the methods is obviously higher than with S1V: this is done at the expense of the quality of each variable separately. Hence, in such a case where only one variable is of interest in a spatial context,

the S1V methods have to be favoured. Moreover, generally, the bias reduction in dependence is stronger (i.e., better) for the cross-auto-correlations between precipitation and temperature (matrices "pr/tas") and between precipitation and precipitation (matrices "pr/pr") than for the other couples of variables ("tas/tas" and "tas/pr"). This comes from the fact that the initial biases (i.e., Wasserstein distances) of the raw GCM data for pr/tas and pr/pr are larger than those from tas/tas and tas/pr. This is visible in Fig. S7 given as supplementary material, showing the values of the Wasserstein distances (based on the second

normalization of the DCP sets) used to compute $BR_{\mathcal{W}}$. In addition, some negative $BR_{\mathcal{W}}$ values in Fig. 9 are related to very small differences between Wasserstein distances very close to zero. Indeed, if the model simulations to be corrected have a DCP set already close to the reference, its Wasserstein distance will be near zero. If a corrected dataset has also its distance to the reference close to zero, the resulting ratio can be quite different from 1 (and thus induce a strongly negative $BR_{\mathcal{W}}$ value) while the two $\mathcal{W}$ values are only slightly different. For example among others, this is the case for TSMBC-5 (S1V) at lag 1

for the "tas/tas" matrix under projection context in Fig. S7, where the $\mathcal{W}$ value is small but slightly higher than that of the raw IPSL simulations, implying a negative (red) $BR_{\mathcal{W}}$ value in Fig. 9. Finally, when comparing TSMBC-5 and TSMBC-10, it appears that, whatever the configuration, increasing the dimension (i.e., 10 lags to be accounted for in the MBC, instead of 5) degrades the gain in correlations. The increase in the complexity (i.e., the number of dimensions) of the method is made at the expense of the quality of the results. This was not visible on previous Fig. 8 that only accounted for the shape of the

DCPs. Thus, it indicates that this degradation is mostly due to biases in the marginals that are not fully removed by TSMBC in a high-dimensional setting. One potential explanation for this is the well-known problem of "curse of dimensionality" (e.g., Wilcox, 1961; Finney, 1977): having 2500 values in 4576 dimensions for TSMBC10 / S2V indicates that we may not have enough data to explore such a high-dimensional space and, thus, that the MBC inference/procedure performed by dOTC may not be robust. In addition, an increased number of dimensions could potentially lead to two types of linear dependencies

that could interfere with the underlying MBC method being used (dOTC): (i) a linear dependence between two "close" grid points (especially for temperature), although this effect seems limited as dOTC performed correctly at lag 0; and (ii) a linear

dependence in the lagged matrix by duplicating and shifting the columns. However, the latter is difficult to distinguish from the curse of dimensionality problem.

## 5    Conclusions and discussion

The goal of bias correction (BC) is to transform biased climate simulations in order to make their statistical properties more similar to those from reference data. Over the last decades, many univariate BC methods were developed and applied, working on one climate variable at a time and one location at a time. Over the last few years, various multivariate bias correction (MBC) methods were also designed to correct not only some marginal properties of the simulations (e.g., means, variances, distributions) but also their dependencies (e.g., correlations), either in a multivariate context, inter-site context, or both. Some methods

were even specifically developed to adjust the temporal properties (e.g., Mehrotra and Sharma, 2016) of the simulations. However, the latter usually consider time-related properties and temporal (auto- or cross-) correlations specifically, i.e., differently from inter-variable or inter-site dependencies. The goal of the present study was then to investigate if by considering time just like other variables – i.e., by adding to the (multivariate) data to be corrected some lagged time series of itself –, an MBC method is able to correct both the multivariate (inter-variable and inter-site) properties and the temporal properties. To test this,

the "dynamical Optimal Transport Correction" (dOTC) method has been applied first to a synthetic (i.e., statistically generated) dataset and, second, to a dataset of daily temperature and precipitation over South of France, simulated from the IPSL global climate model. For evaluation, the reference dataset was extracted from higher-resolution regional climate simulations over the same region. dOTC was then applied to correct the IPSL dataset where various lagged versions of those simulations have been added. This approach – performing an MBC on lagged time series in addition to the initial ones – has been called "Time

Shifted Multivariate Bias Correction" (TSMBC). Furthermore, it has been tested based on 4 configurations: only one variable is corrected at a time, either for a given location (L1V) or for all locations at the same time (S1V); the 2 variables are jointly corrected, for a single location (L2V) or for all locations (S2V).

From the synthetic data experiment, a comparison with a "reasonably naive" multivariate bias correction method, RBC, based

on random sampling, has been proposed. The results showed that:

–    TSMBC(dOTC) provides a clear improvement compared to TSMBC(RBC);

–    the choice of the starting row only has a marginal influence on the corrections. In the case of a starting row $n > 1$ (i.e., with a lag $s > 0$), the reconstruction will omit the first $n-1$ time steps. In order to have as many reconstructed time steps as in the model simulations to correct, it is possible to sample from the first $s-1$ row(s) of the fully corrected lagged

matrix, allowing to complete first $n-1$ time steps of the reconstruction matrix. However, as no values are omitted when starting at first row ($r = 1$) for the reconstruction, this is a logical and practical choice;

–    for a relatively low number $s$ of lags to account for in TSMBC — say $s \leq 10$-day —, the TSMBC-$s$ results are roughly equivalent, whatever $s$.

Those first conclusions indicates some robustness of the proposed TSMBC methodology that, despite some choices to make by the user – starting row, number of lags to include –, provides stable corrections.

In order to evaluate the results in a fully multivariate manner (i.e., intervariable, inter-site and temporal aspects), a new statistical criterion has been proposed. It is based on the Wassertein distance between the set of distance/correlation pairs (DCP) from reference and that of a dataset (from corrections or simulations). This distance can be computed on lagged data, multiple variables and at different locations, hence providing assessments of cross-auto-correlations, generalizing the traditional correlogram tool.

The results obtained by applying TSMBC to climate simulations provided the following conclusions:

- In terms of means and standard deviations, for both temperature and precipitation, the inclusion of lagged data does not strongly modify the results of the dOTC correction method. Although some evidences of degradation might appear when the number of lags increase (e.g., for $BR_\sigma$ in temperature), the bias reduction values with respect to the raw simulations are mostly positive, indicating an improvement over the raw GCM.

- This is also mostly the case for autocorrelation bias reductions $BR_\rho$ but, this time, the increase in number of lags in TSMBC globally improves the results, even though TSMBC-10 does not clearly improve TSMBC-5.

- Moreover, the main spatio-temporal patterns of the TSMBC results are globally improving those from the raw GCM.

- However, biases in the intensities of the (intervariable, inter-site or temporal) correlations might remain. This is typically related to very small differences between two Wasserstein distances very close to zero: if the raw simulations already have a DCP set close to the reference, its Wasserstein distance will be near zero. Therefore, the relative reduction of bias $BR$ can be strongly negative, even though the absolute difference is potentially very small.

- Finally, if the TSMBC methodology seems to reasonably adjust temporal (cross-auto) correlations, while still performing well on multivariate properties, when the number of lags increases (e.g., from 5 to 10 days), the gain in the quality of the corrections is not obvious and the latter can even be degraded. It is thus required to limit the temporal constrains to a few time steps, depending on the variable of interest. This would avoid to apply the MBC method in a too high dimensional context and then allow robust results.

Globally, the results of the different tests indicate that the proposed approach of "time shifted multivariate bias correction" – i.e., including lagged versions of the simulated and reference datasets in a multivariate correction procedure – can indeed be relevant to adjust temporal properties, in addition to more usual marginal and multivariate components.

Despite its promising results, the TSMBC approach can be further investigated and improved. For example, in the present study, only the dOTC multivariate method was used as correction technique. Other MBC methods exist. Hence, it would be interesting to test how those alternative MBCs – such as "$R^2D^2$" (Vrac, 2018; Vrac and Thao, 2020), "MBCn" (Cannon, 2018), or "MRrec" (Bárdossy and Pegram, 2012), among others – would behave in this TSMBC framework.

Note also that the chosen lag in TSMBC should be adapted to the type of variable and the area. For example, taking 3 days ($s = 3$) for precipitation in Europe seems reasonable, while pressure or temperature could require a week ($s \geq 7$). Hence, a preliminary analysis of the autocorrelation or temporal properties of the variables to be corrected should be performed to decide about the relevant lag to use.

In addition, when dealing with precipitation, the rainfall occurrence is not treated differently from the non-occurrence (dry days) by the TSMBC approach proposed here (i.e., using dOTC as underlying MBC method). However, the sequences of dry days and wet days can bear a major part of the autocorrelation information. Hence, it could be interesting to account for this specific aspect of precipitation when performing the underlying MBC method.

Moreover, some adjustment methods were designed to account specifically for the correction of temporal properties (e.g., Johnson and Sharma, 2012; Mehrotra and Sharma, 2015, 2016). A comparison of TSMBC to such methods would then be of interest to understand if specific modelling is needed for adjusting temporal properties or if considering lagged data into MBCs (i.e., no specific modelling as in TSMBC) provides equivalent results.

Finally, the Wasserstein cross-auto-correlation based metric introduced in this study could be used more generally to compare various datasets and/or assess their diverse properties with respect to a reference. It can then be useful to make evaluations of climate simulations (adjusted or not) in a more holistic way.

*Code and data availability.* The CMIP5 and CORDEX database are freely available. Source codes of TSMBC are freely available in the R/Python package SBCK under the GNU-GPL3 license (Robin, 2021).

## 530   Appendix A: Optimal transport and the dOTC method

A bias correction method is classically defined as a map $\mathcal{T}$ between the biased dataset, noted $\mathbf{X}$ with probability distribution $\mathbb{P}_{\mathbf{X}}$ on $\mathbb{R}^d$, and the reference dataset, noted $\mathbf{Y}$ with probability distribution $\mathbb{P}_{\mathbf{Y}}$ on $\mathbb{R}^d$; such that:

$$\mathcal{T}(\mathbb{P}_{\mathbf{X}}) = \mathbb{P}_{\mathbf{Y}}$$

Consequently, a biased value $\boldsymbol{x} \in \mathbf{X}$ is linked to its correction $\boldsymbol{y} \in \mathbf{Y}$ by the relation $\boldsymbol{y} = \mathcal{T}(\boldsymbol{x})$. Robin et al. (2019) have
replaced the map $\mathcal{T}$ by a probability distribution $\gamma$ on $\mathbb{R}^d \times \mathbb{R}^d$, such that $\gamma(\boldsymbol{x}, \boldsymbol{y})$ *is the probability that $\boldsymbol{y}$ is the correction of $\boldsymbol{x}$*. The case of a map $\mathcal{T}$ corresponds to the probability distribution defined on the couples $(\boldsymbol{x}, \mathcal{T}(\boldsymbol{x}))$. The set of probability distribution $\gamma$ (or bias correction methods) is noted $\Gamma$, and given by:

$$\Gamma = \left\{ \gamma \; : \; \mathbb{P}_{\mathbf{X}}(A) = \int_{A \times \mathbb{R}^d} \mathrm{d}\gamma, \; \mathbb{P}_{\mathbf{Y}}(B) = \int_{\mathbb{R}^d \times B} \mathrm{d}\gamma \right\}.$$

With this formulation, $\mathbb{P}_{\mathbf{X}}$ and $\mathbb{P}_{\mathbf{Y}}$ are the first and second projection of all $\gamma \in \Gamma$. This set is non-empty, because it contains
$\gamma_{\mathrm{RBC}} = \mathbb{P}_{\mathbf{X}} \times \mathbb{P}_{\mathbf{Y}}$. The RBC (Random Bias Correction) procedure for $\gamma_{\mathrm{RBC}}$ consists to draw randomly according to $\mathbb{P}_{\mathbf{Y}}$ the correction $\boldsymbol{y}$ of $\boldsymbol{x}$. The dOTC method uses a specific $\gamma$, which minimizes the *energy*.

The dOTC method is given by the $\tilde{\gamma} \in \Gamma$ which minimizes the energy needed to transform $\mathbf{X}$ to $\mathbf{Y}$. It is defined by the minimum of the following cost function, coming from optimal transport theory (see, e.g. Villani, 2008; Santambrogio, 2015):

$$\tilde{\gamma} = \arg\min_{\gamma \in \Gamma} \int_{\mathbb{R}^d \times \mathbb{R}^d} \|\boldsymbol{x} - \boldsymbol{y}\|^2 \mathrm{d}\gamma(\boldsymbol{x}, \boldsymbol{y}) \tag{A1}$$

Here, $\|\boldsymbol{x} - \boldsymbol{y}\|^2$ is the energy needed to transform $\boldsymbol{x}$ to $\boldsymbol{y}$, *weighted by $\gamma(\boldsymbol{x}, \boldsymbol{y})$*. In the univariate case ($d = 1$), $\tilde{\gamma}$ corresponds to the quantile mapping method. To take into account of a projection period, where references are not available, the following modification had been proposed by Robin et al. (2019): noting $\mathbf{X}^C$ and $\mathbf{Y}^C$ the biased and reference dataset in the calibration period, and $\mathbf{X}^P$ the biased dataset in the projection period, two transformations are inferred:

– $\tilde{\gamma} : \mathbf{X}^C \mapsto \mathbf{Y}^C$, the bias, and,

    – $\tilde{\varphi} : \mathbf{X}^C \mapsto \mathbf{X}^P$, the evolution of the biased data.

The correction $\mathbf{Z}^P$ during projection period by dOTC is formally (see Robin et al., 2019, for details) given by:

$$\mathbf{Z}^P = (\tilde{\varphi} \circ \tilde{\gamma} \circ \tilde{\varphi}^{-1})(\mathbf{X}^P).$$

The "evolution" distribution is transferred along the "bias" distribution to construct a correction in the reference world with an evolution similar to that of the biased data. Note that this method seeks to preserve the evolution of the model while reducing bias. This idea is similar to the CDF-t method (see, e.g. Vrac et al., 2012), which extends the quantile mapping in a non-stationary context.

## Appendix B: Wasserstein metric

From the probability distribution $\tilde{\gamma}$ defined by the Eq. (A1), a metric, called the *Wasserstein distance*, can be derived. This metric is defined by:

$$\mathcal{W}(\mathbb{P}_{\mathbf{X}}, \mathbb{P}_{\mathbf{Y}})^2 = \int_{\mathbb{R}^d \times \mathbb{R}^d} \|\boldsymbol{x} - \boldsymbol{y}\|^2 \mathrm{d}\tilde{\gamma}(\boldsymbol{x}, \boldsymbol{y}) = \inf_{\gamma \in \Gamma} \int_{\mathbb{R}^d \times \mathbb{R}^d} \|\boldsymbol{x} - \boldsymbol{y}\|^2 \mathrm{d}\gamma(\boldsymbol{x}, \boldsymbol{y}) \tag{B1}$$

The Wasserstein metric is sensitive to the shape of the distribution, and is a measure of "how much is the cost to transform $\mathbb{P}_{\mathbf{X}}$ to $\mathbb{P}_{\mathbf{Y}}$". Hence, a value of $\mathcal{W}(\mathbb{P}_{\mathbf{X}}, \mathbb{P}_{\mathbf{Y}})$ close to 0 indicates that the two (multivariate) distributions $\mathbb{P}_{\mathbf{X}}$ and $\mathbb{P}_{\mathbf{X}}$ are similar, while a large value indicates that the distributions are different.

*Author contributions.* MV and YR had the idea of the method together. They designed the study and the experiments together. YR made the computations and plots. YR and MV jointly analysed the results. MV wrote the major part of the article with inputs from YR.

*Competing interests.* The authors declare no competing interest.

*Acknowledgements.* We acknowledge the World Climate Research Program's Working Group on Coupled Modelling, which is responsible for CMIP, and we thank the climate modeling groups for producing and making available their model output. For CMIP the U.S. Department of Energy's Program for Climate Model Diagnosis and Intercomparison provides coordinating support and led development of software infrastructure in partnership with the Global Organization for Earth System Science Portals.

We acknowledge the World Climate Research Programme's Working Group on Regional Climate, and the Working Group on Coupled Modelling, former coordinating body of CORDEX and responsible panel for CMIP5. We also thank the climate modelling groups for producing and making available their model output. We also acknowledge the Earth System Grid Federation infrastructure an international effort led

by the U.S. Department of Energy's Program for Climate Model Diagnosis and Intercomparison, the European Network for Earth System Modelling and other partners in the Global Organisation for Earth System Science Portals (GO-ESSP).

580    MV has been supported by the CoCliServ project. MV and YR have been supported by the EUPHEME project. Both CoCliServ and EUPHEME are part of ERA4CS, an ERA-NET initiated by JPI Climate and cofunded by the European Union (grant no. 690462). MV has also been supported by project C3S 428J ("HR-CDFt"). YR has also been supported by project C3S 62 ("Prototype Extreme Events and Attribution Service").

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

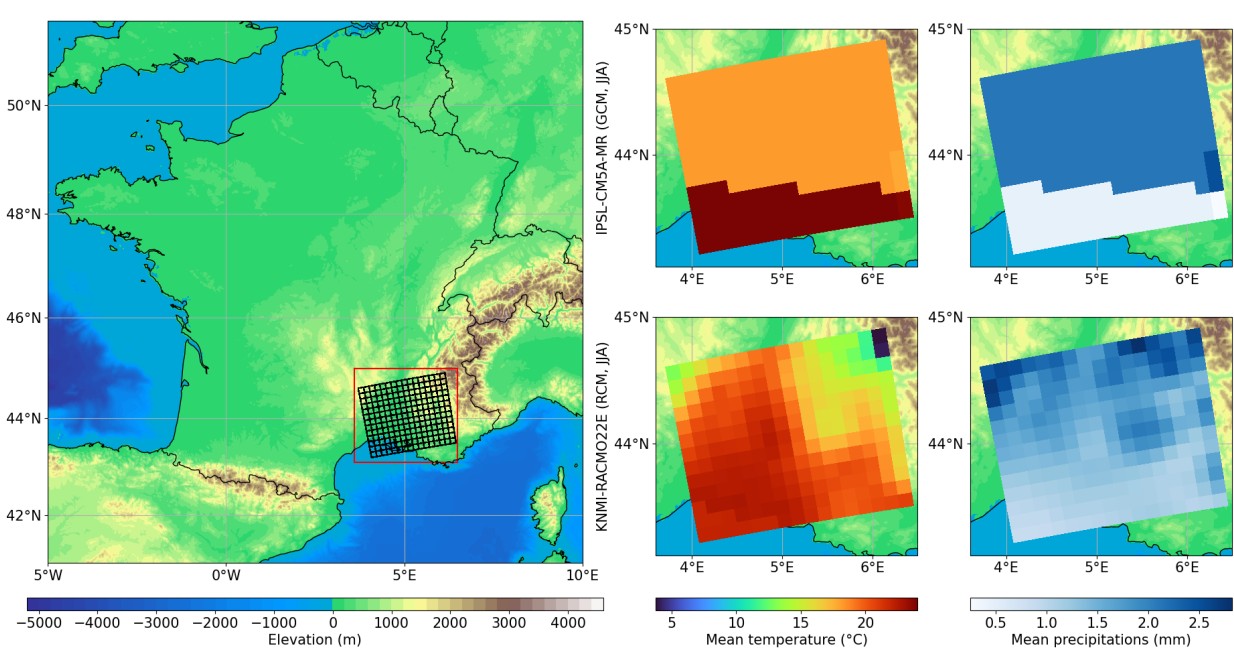

**Figure 1. Left:** Map of elevation (in m) over France. The region of interest lies in the south-east box. **Right:** Mean summer (JJA) temperature (in $^{o}C$) and precipitation (in mm/day) for the region of interest (up: IPSL-CM5A-MR; bottom: KNMI-RACMO22E) over 1951-2010.

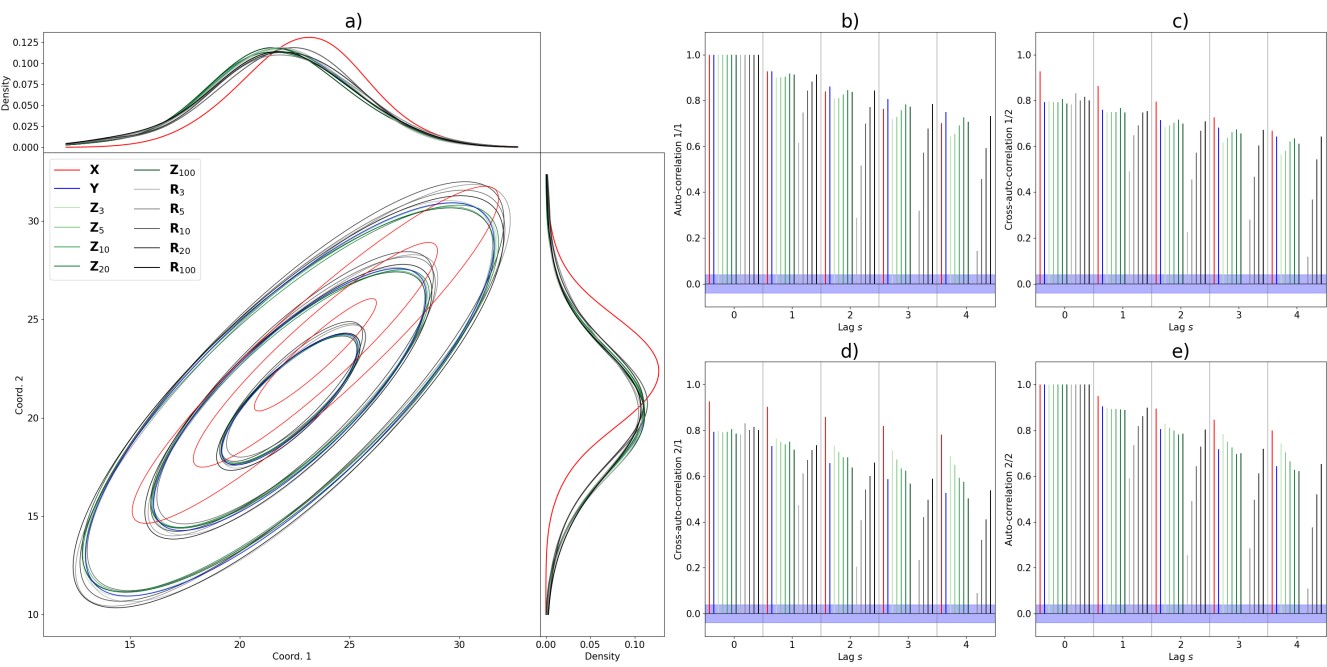

**Figure 2. a)** Probability densities and covariance matrices of the VAR processes and their corrections. **b)** Auto-correlation of the first coordinate. **c)** Cross-auto-correlation between the first and the second coordinates. **d)** Cross-auto-correlation between the second and the first coordinates. **e)** Auto-correlation of the second coordinate.

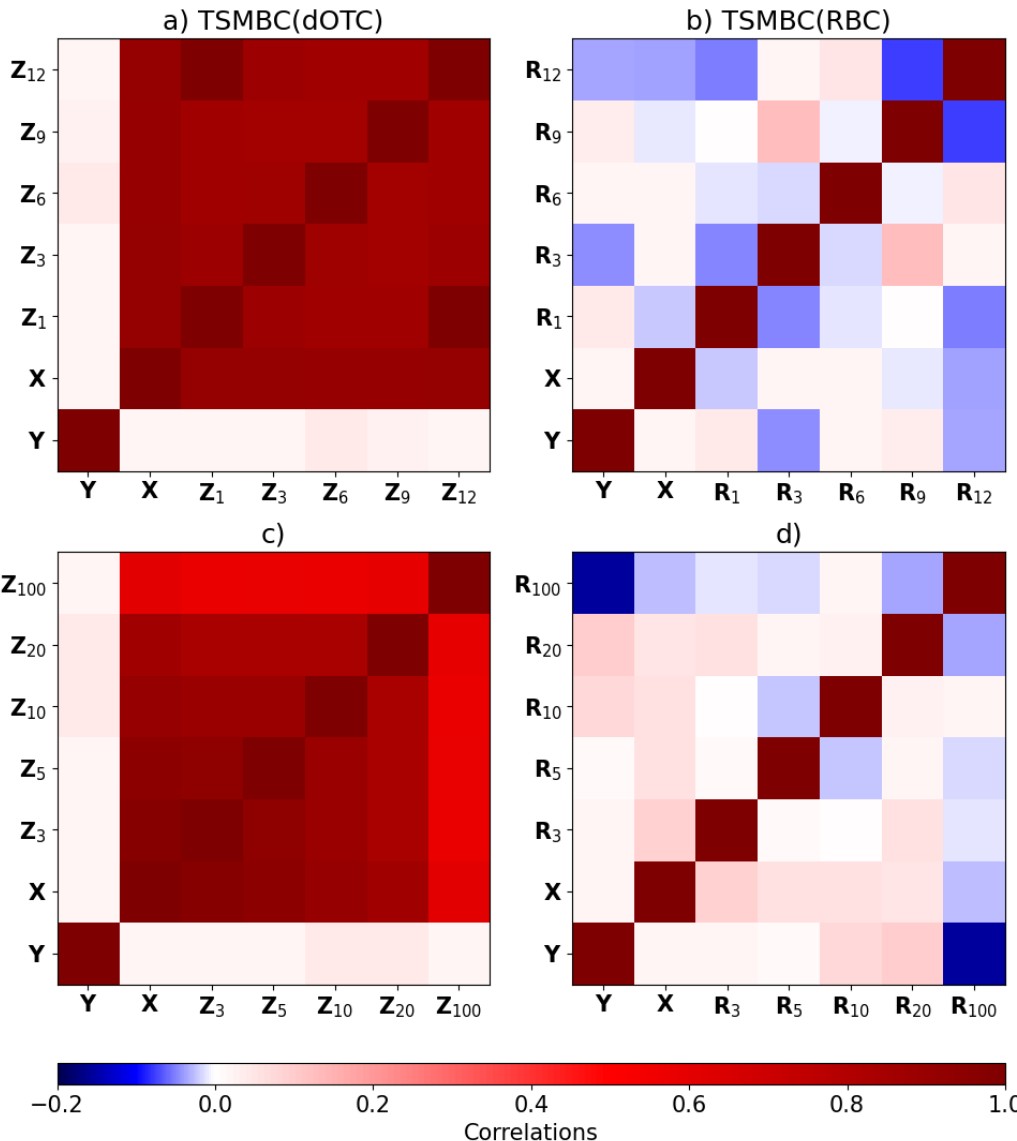

**Figure 3.** The first row of matrices is the correlations between corrections of the VAR with different choice of starting row for the reconstruction step. The second row is the correlations between corrections of the VAR with different choice of lag. The starting row for the reconstruction step is the middle value between 0 and lag of corrections. The first column is the correction with the method TSMBC with the underlying method dOTC, and the second column is also TSMBC, but with the underlying method RBC.

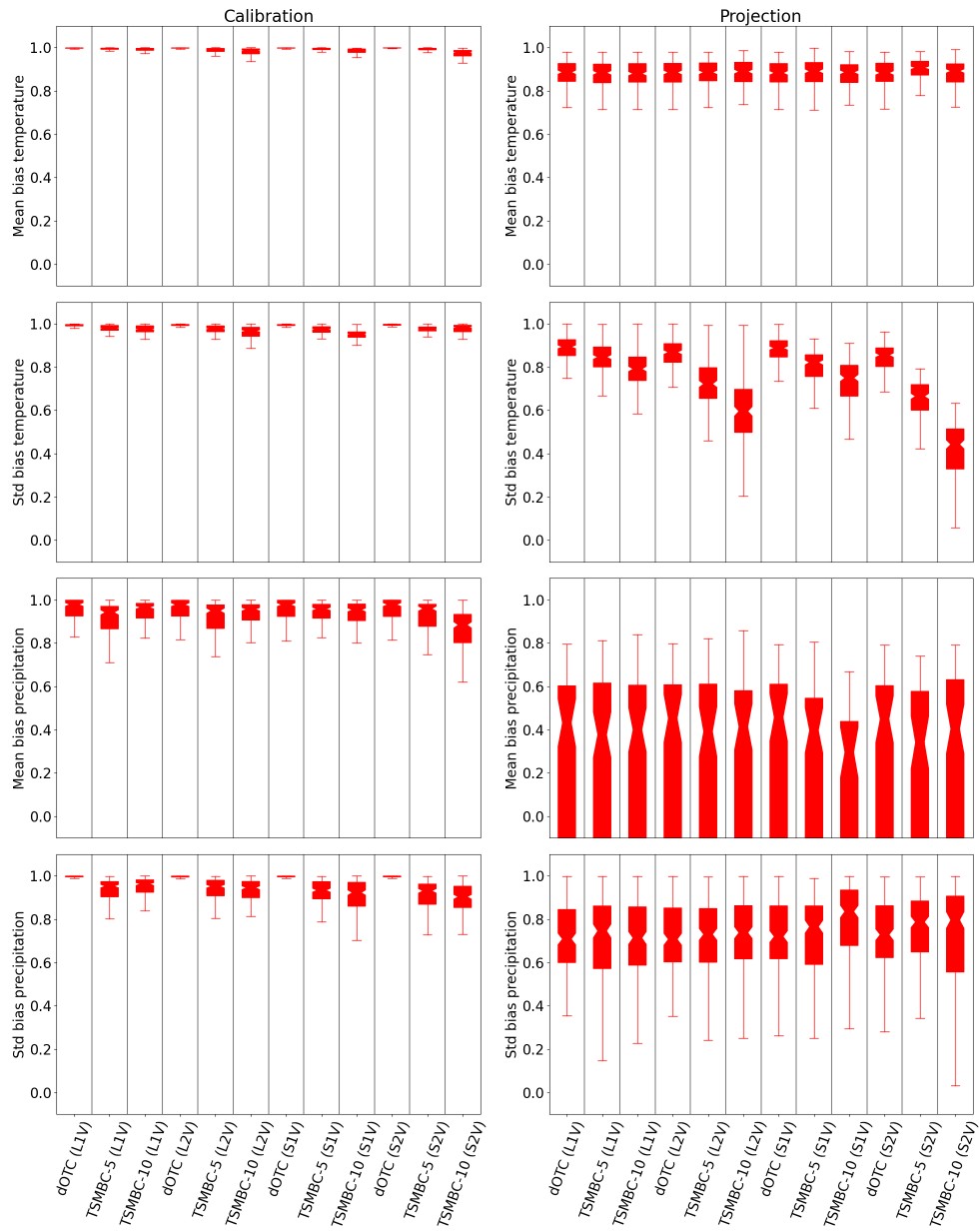

**Figure 4.** Boxplots of bias reduction in mean and standard deviation ($BR_{\mathbb{E}}$ and $BR_{\sigma}$) in calibration (first column) and projection (second column) periods for temperature and precipitations in summer. The closer the boxplot is to 1, the closer the results are to the reference and therefore the better they are.

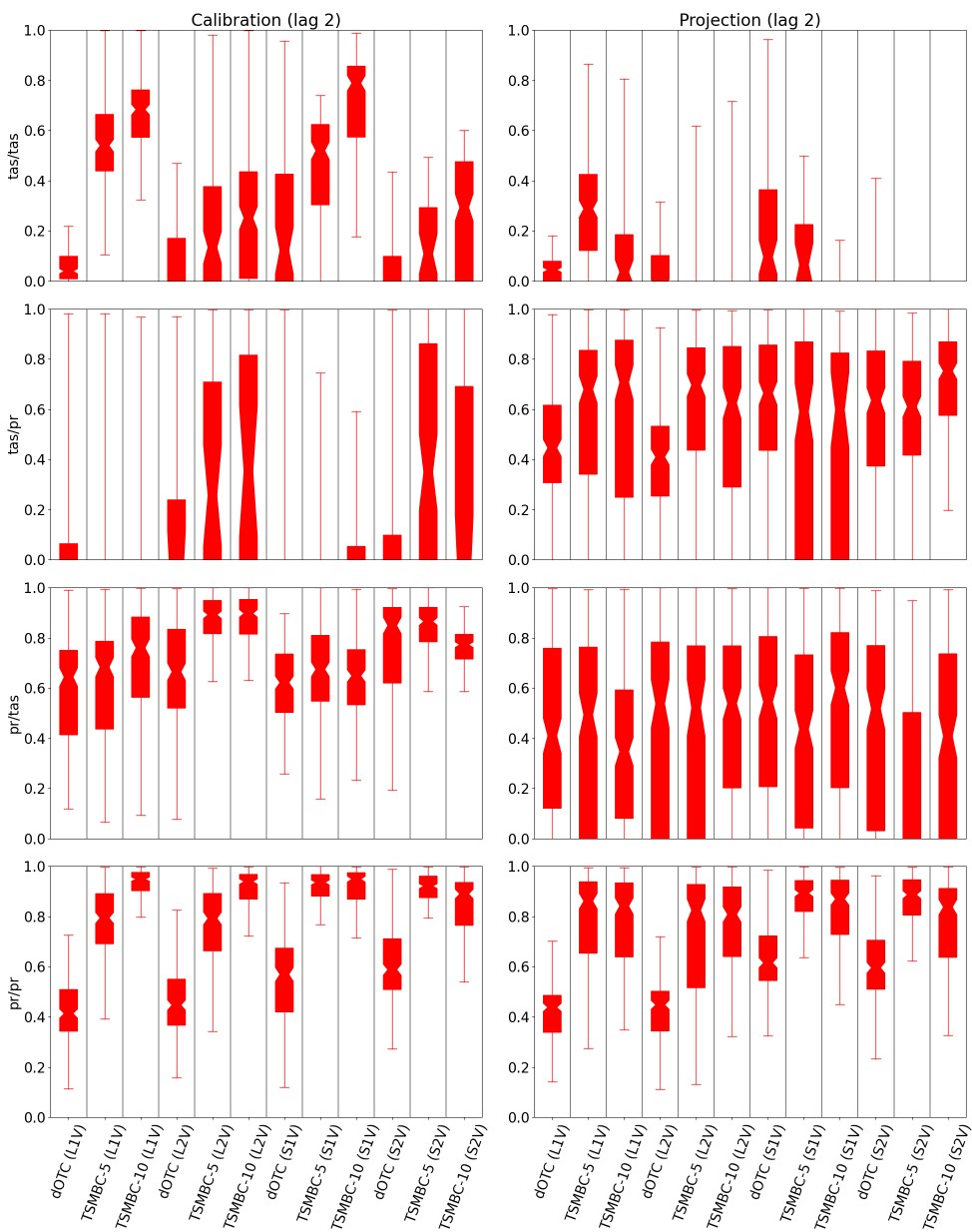

**Figure 5.** Boxplots of bias reduction in lag 2-(cross)-auto-correlations ($BR_{\rho_2}$) in calibration (first column) and projection (second column) period for temperature and precipitations in summer. The closer the boxplot is to 1, the closer the results are to the reference and therefore the better they are.

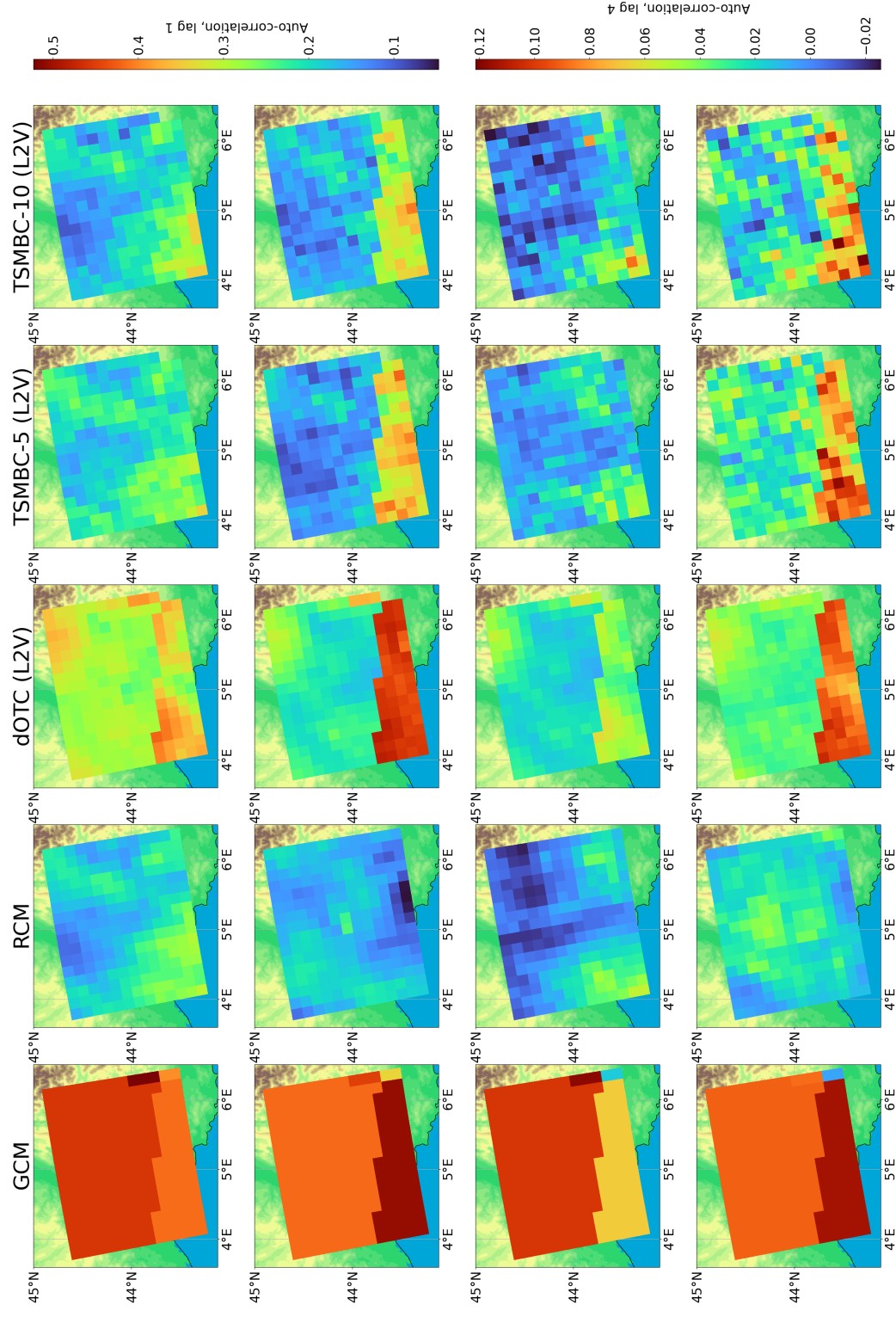

**Figure 6.** Map of lag-1 (two first rows) and lag-4 (two last rows) auto-correlations of precipitation in summer. The first and third rows are for the calibration period, the second and fourth rows are for the projection period. The first (resp. second, third, fourth and last) column gives the maps of auto-correlations of the RCM to correct (resp. the GCM, the correction dOTC, the correction TSMBC-5 and the correction TSMBC-10 with method L2V). A key point of this figure is to compare the evolution between calibration and projection periods for the RCM, GCM and the corrections. The evolution of the TSMBC corrections is similar to the GCM evolution, which is different from the RCM evolution, leading to a fail of the correction in projection period.

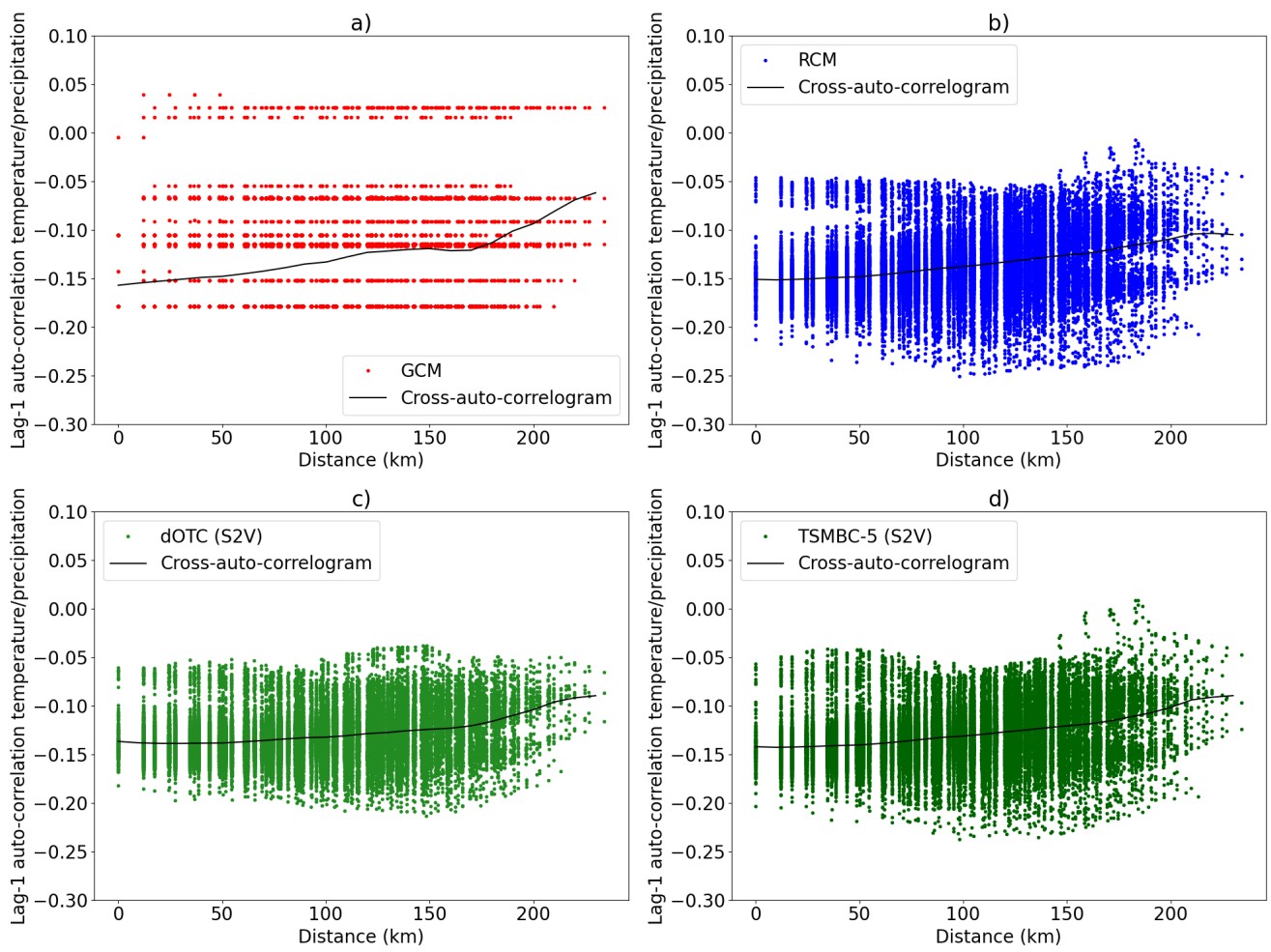

**Figure 7.** Auto-correlogram (lag 1) between temperatures and precipitation in calibration period in summer for **a)** the model; **b)** the observations; **c)** the S2V dOTC correction and **d)** the S2V TSMBC-5 correction.

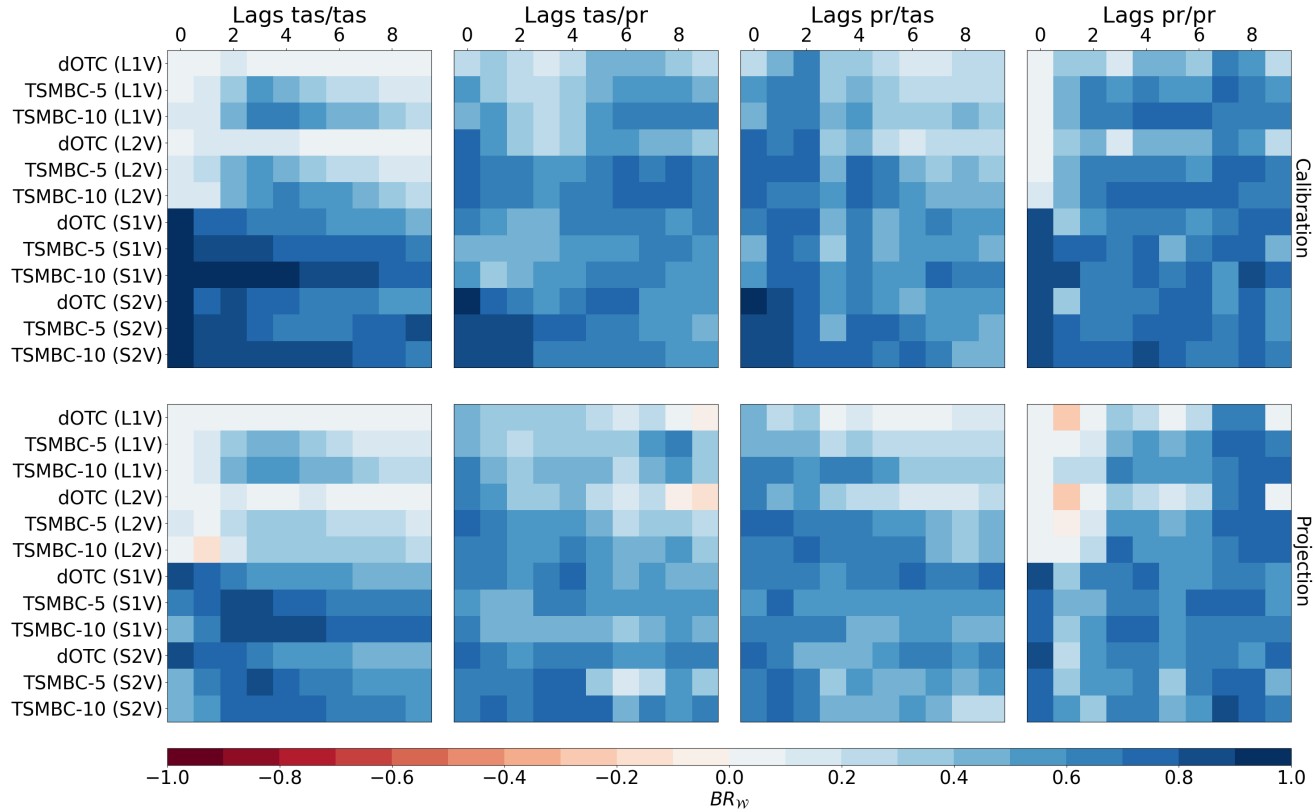

**Figure 8.** Bias reduction of dependence ($BR_{\mathcal{W}}$) values, based on the Wasserstein distances computed on bivariate (correlation, distance) distributions between reference and the different BC datasets or model simulations in summer. The (correlation, distance) 2d-distributions come from the calculated correlograms (see figure 7 and text for details). The first line of matrices corresponds to the calibration period, and the second line to the projection period. The BC results and model simulations correspond to rows. The correlations are calculated between tas and tas (first matrix), tas and pr (second matrix), pr and tas (third matrix) and pr and pr (fourth matrix). For each matrix, the columns correspond to different lags and thus correlations indicate auto-correlations. Hence, the two central matrices (tas/pr and pr/tas) contain cross-correlations and cross-auto-correlations. In order to compare the shape of (correlation,distance) set, a normalization is performed separately for each cell (i.e., each couple method-lag) of each matrix. This normalization allows us characterizing the pattern of the distributions, and to get rid of the marginal properties. Hence, the comparison between different couples method-lag is possible but only to characterize the shape of the (distance, correlation) distributions. The closer the $BR_{\mathcal{W}}$ value is to 1, the closer the results are to the reference and therefore the better they are.

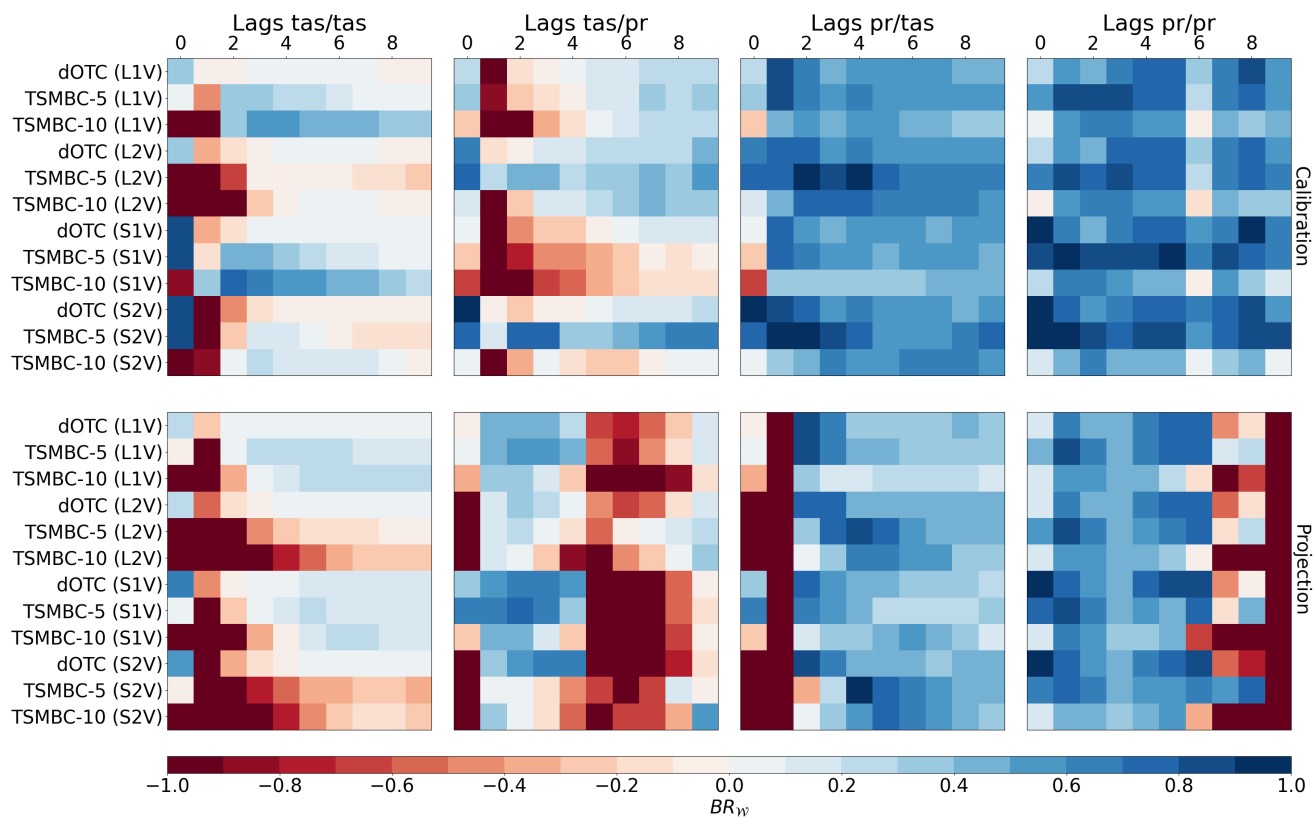

**Figure 9.** Same as Fig. 8, but with a normalization by column (i.e., by given lag), before computing the $BR_\mathcal{W}$ values. Hence, the $BR_\mathcal{W}$ values of a given method for two different lags cannot be compared with this normalization. The closer the $BR_\mathcal{W}$ value is to 1, the closer the results are to the reference and therefore the better they are.

|      | dOTC | TSMBC-5 | TSMBC-10 |
| ---- | ---- | ------- | -------- |
| L1V | $1 \times 1 \times 1 \times 1 = 1$ | $1 \times 1 \times 1 \times (5+1) = 6$ | $1 \times 1 \times 1 \times (10+1) = 11$ |
| L2V | $2 \times 1 \times 1 \times 1 = 2$ | $2 \times 1 \times 1 \times (5+1) = 12$ | $2 \times 1 \times 1 \times (10+1) = 22$ |
| S1V | $1 \times 16 \times 13 \times 1 = 208$ | $1 \times 16 \times 13 \times (5+1) = 1248$ | $1 \times 16 \times 13 \times (10+1) = 2288$ |
| S2V | $2 \times 16 \times 13 \times 1 = 416$ | $2 \times 16 \times 13 \times (5+1) = 2496$ | $2 \times 16 \times 13 \times (10+1) = 4576$ |

**Table 1.** Summary of the dimensions of the bias correction for each methods used in Sec. 4. For each cell, the first value corresponds to the dependence or not between temperature and precipitations. The second and third are the dimension of the grid of the RCM ($16 \times 13$ grid cells). The last values is the dimension to correct until the lag 0 for the first column, the lag 5 for the second column and the lag 10 for the last column.