# Peer review of "Is time a variable like the others in multivariate statistical downscaling and bias correction?"

_Earth System Dynamics, 2021_

## Author Comment (AC1)

**"Is time a variable like the others in multivariate statistical downscaling and bias correction?", reviewer 1**

Yoann Robin & Mathieu Vrac, https://doi.org/10.5194/esd-2021-12

**General comments**

*In this manuscript, a new method of incorporating the temporal variable into a multivariable bias correction is introduced with sufficient motivations and with a thorough and clear description. This new method is versatile in that it can work with any existing MBC's and this is demonstrated via applying it to dOTC and to a more naive method they call Random Bias Correction. The method is first tested on a synthetic dataset for an explorative tuning of the parameters, then to a real dataset. A few points from the analyses from the real data experiments are unconvincing (this will be touched in the specific comments), but most results are well-supported. A new generalizable metric is introduced for measuring bias reduction relative to some ground-truth dataset but its benefits and shortcomings could be discussed further.*

**General response:**

First, we would like to thank this anonymous reviewer for her/his thorough reading and interesting comments. We tried to take them into account and we provide point-by-point responses below in blue.

**Specific comments**

**Comment 1**

*In section 2.2, the concept of reconstruction by rows is introduced. Reconstruction by rows certainly seem to perform better than reconstruction by columns. It is asserted here that many reconstructions are possible and that these are determined by the "starting row". Starting the $n^{th}$ row for $1 < n < l$ for some lag $l$ omits the first $n-1$ values, which are clearly needed in the final reconstruction. It is possible that those $n-1$ values are repeated more than once in the lagged matrix and a more specific description of how to include these values is needed.*

Response:

We thank the reviewer for this question that allows us to clarify the methodology.

Actually, based on our "reconstruction by rows", it is not possible to repeat values. In the case considering $l$ lags, this method by rows jumps (concatenate) from one given row (e.g., nth row) to another row located $l$+1 rows after (i.e., (n+$l$+1)th row) avoiding to repeat values. However, as noted by the reviewer, starting at row n > 1 will omit the n - 1 first values. Generally, this has no impact on the corrections in terms of their statistical properties as n is usually very small compared to the length of the time series. This is validated by the results of the test performed in section 3.1 with a VAR process. To clarify the reconstruction method by rows and the fact that no repeated values are possible, we have added the following sentence in section 2.2:

*"However, even if the values are repeated in the lagged matrix, no repeated values can appear in the final reconstruction. [...] Moreover, the choice of a starting row $r>1$ omits the first $r-1$ values in the final reconstruction. This leads us to wonder about the influence of the choice of the starting row."*.

Nevertheless, as no values are omitted when starting at first row for the reconstruction, this is a logical and practical choice. This is now clarified in the "Conclusions and discussion" section 5 as follows:

*"In the case of a starting row $n>1$ (i.e., with a lag $s>0$), the reconstruction will omit the first $n-1$ time steps. In order to have as many reconstructed time steps as in the model simulations to correct, it is possible to sample from the first $s-1$ row(s) of the fully corrected lagged matrix, allowing to complete first $n-1$ time steps of the reconstruction matrix. However, as no values are omitted when starting at first row ($r=1$) for the reconstruction, this is a logical and practical choice"*.

**Comment 2**

*Section 3.1 asserts that the starting row has little impact on the overall bias correction performance, and this is attributed to the high correlations of the results of the TSMBC method to the biased data matrix X, as well as the high correlations between the results of the TSMBC methods with varying starting rows (as shown in Figure 3). In figure 3, it is also shown that all TSMBC results have very low correlations with Y, the reference matrix. Shouldn't the results of TSMBC be "corrected" and therefore aspire to exhibit higher correlations with Y more than X?*

Response:

Regarding the conclusion that the choice of the starting row has little impact on the BC performance: we do NOT conclude this because of the high correlations of the TSMBC results to the biased data matrix X. We conclude this from the high correlations between the results obtained from different starting rows. Indeed, this indicates that, whatever the chosen starting row, the results are very close to each other. The resulting high correlations with data matrix X is an effect of the dOTC method, which tries to preserve as much as possible the temporal properties of the model simulations to be corrected (Robin et al., 2019; François et al., 2020).

This is now clarified in section 3.1, as follows:

"*We can see for TSMBC(dOTC) that all corrections are highly correlated between them -- with values close to 1 -- whereas for TSMBC(RBC) no significant correlation appears. This indicates that, whatever the chosen starting row for TSMBC, the results are very close to each other. Remark that for TSMBC(dOTC) the corrections stay highly correlated with $\mathbf{X}$. It is an effect of the dOTC method, which tries to preserve as much as possible the temporal properties of the model simulations to be corrected(Robin et al., 2019; François et al., 2020).*"

Regarding whether or not the TSMBC results should aspire to exhibit higher correlations with Y than with X: Actually, the answer is "no". The goal of bias correction, be it univariate or multivariate and with or without including a correction of the temporal properties, is to have corrected simulations that have statistical properties similar to those from the reference data. Hence, the goal is not to get corrected data correlated to observations. Raw climate simulations and references are usually uncorrelated (except via seasonal cycles and potential trends). As some bias correction methods preserve the rank chronology (i.e., the temporal properties) of the simulations --- as is the case, to some extent, for the dOTC method (see Robin et al., 2019) ---, there is no reason for the bias corrected data to be correlated to the reference. In other words, if the BC procedure is efficient, corrected and reference time series can be seen as generated based on the same statistical distributions and/or properties but independently. Hence, they are not correlated.

Clarifications have been brought to the manuscript by adding the following text into section 3.1:

"*On the other hand, no correlation appears with Y. This was expected. Indeed, there is no reason for the bias corrected data to be correlated to the reference. If the BC procedure is efficient, corrected and reference time series can be seen as generated based on the same statistical distributions and/or properties but independently. Hence, they are not correlated.*"

**Comment 3**

*The major aspect of TSMBC is that by adding lagged versions of the original time series data, the data is augmented to include the temporal variable as just another variable. This initial mapping from a dimension of size $N_X\times d$ to $(N_X-s)\times d(s+1)$ is injective but the inverse mapping is not. The authors chose to use a simple reconstruction that only relies on one extra parameter, the starting row, as a way to choose this inverse mapping, and assert in section 3.1 that the choice of the starting row does not have a big impact. Given that the analysis of figure 3 is unconvincing, it may be important to more carefully consider how to design the inverse mapping. For example, what is the variance of the repeated values? For TSMBC with lag $s$, there are some time indices that are repeated $s+1$ times total in the reconstruction. Are those $s+1$ values all very close to each other? If not, should some averaging scheme be used? If not, what does the variability in the reconstruction at some time index indicate about whether it should be trusted?*

Response:

It is true that the initial mapping is injective and that, in general, the inverse mapping is not. However, with the suggested reconstruction (i.e., inverse mapping), when a starting row is given, the inverse mapping gives a unique time series and is thus injective.

This is now explained in section 2.2 of the updated article:

"*It is also worth noting the initial mapping (i.e.,going from $\bf{X}$ to $\bf{M\_X}$ is injective, and that, in general, an inverse mapping is not. However, with the suggested reconstruction (i.e., inverse mapping), when a starting row is chosen, the inverse mapping gives a unique time series and is thus injective.*"

Moreover, as explained in response to comment 1, based on our suggested reconstruction method by row, there are no repeated values. Hence, statistics asked by the reviewer cannot be computed.

**Comment 4**

*Regarding the analysis of figure 8 (pg 13, lines 372-389): The statement in line 374-375 "Generally speaking, for a specific configuration of the method (i.e., L1V, L2V, S1V or S2V), TSMBC (5 or 10) is better than dOTC that does not account for temporal properties. " is not well supported by figure 8. Apart from the plots for tas/tas (first column in figure 8), it is difficult to see that the TSMBC cells show darker (higher BR_w) values than the naive comparison dOTC. In addition, shouldn't the 3 methods (dOTC, TSMBC5, TSMBC10) all show the same value/color for lag 0 for each L1V, L2V, S1V, and S2V? What are some reasons they are not?*

Response:

We thank this reviewer for this precious remark. Indeed, TSMBC (5 or 10) is not always better than dOTC. The text has then been modified to describe the results more precisely in section 4.2:

"*Generally speaking, for the local configurations (L1V and L2V), TSMBC (5 or 10) is better than dOTC that does not account for temporal properties. This is true for almost all lags $>$0 and any $BR_{\mathcal{W}}$ matrix (tas/tas, tas/pr, pr/tas, pr/pr). However, for the spatial configurations (S1V and S2V), TSMBC does not seem to provide better results than dOTC, except for the tas/tas matrix where TSMBC strongly improves dOTC.*"

Note also that we have changed the colormap used to better show the details.

Regarding whether or not the 3 methods (dOTC, TSMBC-5 and TSMBC-10) should give the same results for lag 0: The three methods dOTC, TSMBC-5 and TMSBC-10 works differently and with different variables due to the different configurations. For L1V, dOTC works in a univariate context, TSMBC-5 in a 6-dimensional context and TSMBC-10 in a 11-dim context. The number of variables managed by the 3 methods is different for each configuration and increases up to S2V where dOTC works with 416 variables, TSMBC-5 with 2496 variables and TSMBC-10 with 4576 variables (see Table 1). Thus a variability will necessarily appear in the various corrections. However, we can see in Fig. 8 that, for lag 0, the improvements are of the same order for all 3 methods.

**Comment 5**

*One justification for why TSMBC10 performs worse than TSMBC5 is given by the fact that the inflated data size $(N\_X-10)\times d(10+1)$ results in a higher complexity method. In line 412-413, it is stated "The increase in the complexity (i.e., the number of dimensions) of the method is made at the expense of the quality of the results." This is a vague statement and could be made stronger with more specific ideas. For example, the increased number of dimensions could potentially lead to linear dependence which then could interfere with the underlying MBC method being used. There could be some other ways that the increased complexity could have negative effects, and they should be discussed in more detail. Given the size of the problem, numerical instability should probably be ruled out.*

Response:

We did not want to suggest that the problem was due to a numerical instability, but rather that this is related to the well-known problem of "curse of dimensionality": having ~2500 values in 4576 dimensions for TSMBC10 / S2V indicates that we may not have enough data to explore such a high-dimensional space and, thus, that the MBC inference/procedure performed by dOTC may not be robust. However, even in this TSMBC-10/S2V configuration, the "shape" of the DCP set appears improved (first normalization, Fig. 8) whereas a bias appears in the DCP set when the intensity of the correlations are also accounted for (second normalization, Fig. 9).

Regarding the linear dependency, two kinds of dependency might indeed appear:

- The linear dependence between two "close" grid points (especially for temperatures). However, this effect seems limited, as dOTC works correctly at lag 0.
- The linear dependence in the lagged matrix when duplicating and shifting the columns. However, this is difficult to distinguish from the "curse of dimensionality" problem.

At the end of the section 4.2 we have added the following text:

*"One potential explanation for this is the well-known problem of "curse of dimensionality" (e.g.,Wilcox, 1961; Finney, 1977): having 2500 values in 4576 dimensions for TSMBC10 / S2V indicates that we may not have enough data to explore such a high-dimensional space and, thus, that the MBC inference/procedure performed by dOTC may not be robust. In addition, an increased number of dimensions could potentially lead to two types of linear dependencies that could interfere with the underlying MBC method being used (dOTC): (i) a linear dependence between two "close" grid points (especially for temperature), although this effect seems limited as dOTC performed correctly at lag0; and (ii) a linear dependence in the lagged matrix by duplicating and shifting the columns. However, the latter is difficult to distinguish from the curse of dimensionality problem."*

Added references:

Finney, D. J.: Dimensions of Statistics, Journal of the Royal Statistical Society: Series C (Applied Statistics), 26, 285–289, https://doi.org/https://doi.org/10.2307/2346969, https://rss.onlinelibrary.wiley.com/doi/abs/10.2307/2346969, 1977.

Wilcox, R. H.: Adaptive control processes—A guided tour, by Richard Bellman, Princeton University Press, Princeton, New Jersey, 1961, 255pp., Naval Research Logistics Quarterly, 8, 315–316, https://doi.org/https://doi.org/10.1002/nav.3800080314, https://onlinelibrary.wiley.com/doi/abs/10.1002/nav.3800080314, 1961.

**Comment 6**

*Regarding the BR_{\Kappa} metric. One downside of this metric is explained well in the conclusions, in line 458-461: "However, biases in the intensities of the (intervariable, inter-site or temporal) correlations might remain. This is typically related to very small differences between two Wasserstein distances very close to zero: if the raw simulations already have a DCP set close to the reference, its Wasserstein distance will be near zero. Therefore, the relative reduction of bias BR can be strongly negative, even though the absolute difference is potentially very small."*

*Maybe this point should be suggested when the metric is first introduced in section 4.1.*

Response:

This point has been added at the beginning of section 4.1, after Equation (6) :

"*Note that if the raw simulations already have a DCP set close to the reference, its Wasserstein distance will be near zero. If the correction gives also a Wasserstein distance very close to zero, then the relative reduction of bias $BR_\kappa$ can have a very strong negative value if $\kappa(\bf{Z}) > \kappa(GCM)$, even if the absolute difference (i.e., $\kappa(\bf{Z}) - \kappa(GCM)$) is potentially very small.*"

Note also that the (previous) appendix A has been replaced by two appendices: the first one (new appendix A) describes how a bias correction method can be considered as a probability distribution and how dOTC works in this context; the second one (new appendix B) describes the Wasserstein metric. These appendices are not cited here for sake of space.

**Technical comments**

1. *Should "corrected" in line 227 be "correlated" instead?*
2. *Line 289 should have [-\infty,1] instead of ]-\infty, 1]*

Response:

These two technical comments have been corrected.

---

## Author Comment (AC2)

**"Is time a variable like the others in multivariate statistical downscaling and bias correction?", reviewer 2**

Yoann Robin & Mathieu Vrac, https://doi.org/10.5194/esd-2021-12

**General comments**

*This manuscript deals with a new approach (TSMBC) of how to incorporate the time as additional variable into a multivariable bias correction. The approach can be conducted with existing multivariate BC methods such as MBCn, R2D2, or MRrec. Here, the dOTC approach is followed and the results are compared to a "naive" method, the "Random Bias Correction" (RBC).*

*The method is first tested on a synthetic dataset, following a VAR process, before applying it to "real" climate data, based on a pseudo reality approach, i.e. treating the RCM results as observations.*

*The approach could potentially be interesting and innovative. It seems that this is the first time that the time is treated as separate variable in the bias correction. However, I have some doubts that the results are reliable for the application with the real data case (see detailed comments below). Moreover, I think that the evaluation of the TSMBC using synthetic data based on the VAR process is of limited value. It did not convince me technically and scientifically, nor did it help me to better understand the proposed procedure.*

*On the other hand, more information is required to understand the potential value of the TSMBC. Authors did not convincingly present the methodological background. Critical questions remain unanswered, e.g. what is a VAR process? How is the sampling from the VAR process done? How does the dOTC works?*

*The Wasserstein metric is also not well introduced in the method section.*

**General response:**

First, we would like to thank this anonymous reviewer for her/his thorough reading and interesting comments. We tried to take them into account and we provide point-by-point responses below in blue.
More specifically, questions/remarks regarding synthetic data generated with the VAR process, the dOTC methodology, the Wasserstein metric and the naive RBC method are treated in the responses to comment 6 of the reviewer.

**Specific comments**

**Comment 1**

*It remains spurious how and why the increase of the numbers of dimensions (could be time lags or other "variables") affects the stability of the approach. It is just mentioned that the dimensionality should not exceed 10.*

Response:

This is related to the well-known problem of "curse of dimensionality": having ~2500 values in 4576 dimensions for TSMBC10 / S2V indicates that we may not have enough data to explore such a high-dimensional space and, thus, that the MBC inference/procedure performed by dOTC may not be robust. However, even in this TSMBC-10/S2V configuration, the "shape" of the DCP set appears improved (first normalization, Fig. 8) whereas a bias appears in the DCP set when the intensity of the correlations are also accounted for (second normalization, Fig. 9).

Another potential explanation is also the linear dependencies arising from our datasets. Two kinds of linear dependency might appear:

- The linear dependence between two "close" grid points (especially for temperatures). However, this effect seems limited, as dOTC works correctly at lag 0.
- The linear dependence in the lagged matrix when duplicating and shifting the columns. However, this is difficult to distinguish from the "curse of dimensionality" problem.

At the end of the section 4.2 we have added the following text:

*"One potential explanation for this is the well-known problem of "curse of dimensionality" (e.g.,Wilcox, 1961; Finney, 1977): having 2500 values in 4576 dimensions for TSMBC10 / S2V indicates that we may not have enough data to explore such a high-dimensional space and, thus, that the MBC inference/procedure performed by dOTC may not be robust. In addition, an increased number of dimensions could potentially lead to two types of linear dependencies that could interfere with the underlying MBC method being used (dOTC): (i) a linear dependence between two "close" grid points (especially for temperature), although this effect seems limited as dOTC performed correctly at lag0; and (ii) a linear dependence in the lagged matrix by duplicating and shifting the columns. However, the latter is difficult to distinguish from the curse of dimensionality problem."*

Added references:

Finney, D. J.: Dimensions of Statistics, Journal of the Royal Statistical Society: Series C (Applied Statistics), 26, 285–289, https://doi.org/https://doi.org/10.2307/2346969, https://rss.onlinelibrary.wiley.com/doi/abs/10.2307/2346969, 1977.

Wilcox, R. H.: Adaptive control processes—A guided tour, by Richard Bellman, Princeton University Press, Princeton, New Jersey, 1961, 255pp., Naval Research Logistics Quarterly, 8, 315–316, https://doi.org/https://doi.org/10.1002/nav.3800080314, https://onlinelibrary.wiley.com/doi/abs/10.1002/nav.3800080314, 1961.

**Comment 2**

*I have some concerns about applying a BC using climate simulations (based on GCMs and not on reanalysis data) if the temporal sequence of variables is addressed, however, in this case I think it would be acceptable, since the reference is not observation data but downscaled results of the same forcing GCM.*

Response:

Indeed, references are regional climate simulations forced by the same GCM to be downscaled/bias corrected. This kind of "perfect model experiment", considering simulations as "pseudo-observations", is a common approach to assess downscaling / bias correction methods (see, e.g. Charles et al. 2004[1]; Vrac et al., 2007[2], Frost et al. 2011[3]; Bürger et al. 2012[4]; Grouillet et al. 2016[5]).

A clarification has been added in Section 2.1:

"*This kind of "perfect model experiment", i.e., considering simulations as "pseudo-observations", is now a common approach to assess a downscaling / bias correction methods, (see, e.g. Charles et al. 2004; Vrac et al., 2007, Frost et al. 2011; Bürger et al. 2012; Grouillet et al. 2016).*"

**Comment 3**

*My main concern stems from Figure 1 (right, top line). It seems that the mean precipitation and temperature fields do not correspond to the coast line, as I would strongly assume. Due to the coarse resolution, you would expect some distortions in the overlay, but this looks really erroneous. It seems that the projection of GCM and RCM is wrong, it could be reversed left to right.*

Response:

To make sure that we did not make any mistake, we have plotted the equivalent of figure 1 but at the scale of France (see figure at the end of this file). When looking at the maps for France, it is clear that the main geographical patterns are correctly located (e.g., the Alps and the Pyrenees). The patterns visible in figure 1 of the article are also visible here. This shows that Figure 1 is correct.
* * *
[1] https://onlinelibrary.wiley.com/doi/abs/10.1002/hyp.1418
[2] https://doi.org/10.1029/2007GL030295
[3] https://doi.org/10.1016/j.jhydrol.2011.06.021
[4] https://doi.org/10.1175/JCLI-D-11-00408.1
[5] https://doi.org/10.5194/hess-20-1031-2016

**Comment 4**

*Unfortunately, this would have tremendous impacts on the results and interpretations in the following (e.g. the spatial dependencies given in Figure 6). For instance, please explain the statement in lines 300-302. Why is the evolution of GCM variables so different from that of the RCM? Indeed, the RCM includes more spatially-detailed "processes", but is still driven by the GCM. Since the domain of the RCM is rather small, the impact of the forcing is expected to dominate the RCM simulations.*

Response:

The area we have extracted is small, but the boundary of the RCM (i.e., the domain over which the RCM simulations are performed) is Europe, and more precisely the EURO-CORDEX domain, which is much larger than the south-east France domain shown in our article. Consequently, in our domain shown in Fig. 1, the impact of the GCM forcing is not dominating. Also, our domain is framed by 3 mountain ranges (Alps, Pyrenees, Massif Central) badly represented by GCMs while RCMs tend to improve their influences on climate. Hence,the RCM internal dynamics is certainly stronger than that of the GCM.

The following sentence has been added to section 2.1 for clarification:

"*Note that the extracted region of interest is small in comparison to the initial EURO-CORDEX domain (Jacob et al., 2014) over which the RCM simulations were performed.*"

**Comment 5**

*Moreover, I cannot understand the differences the different performances of the calibration and the projection period (Figure 4 & 5). I would expect very similar performances. What is leading to the big discrepancies between the different periods?*

Response:

In a cross-validation setting (i.e., calibration done on a dataset and evaluation/projection performed on a different dataset), the references are not used to fit the model (here, the correction) over the projection period. Therefore, it is expected to have lower quality results over the projection period.

Generally, in a cross-validation within a bias correction context, the quality of the results depends on two elements:

- The ability of the method itself to perform a relevant bias correction;
- The difference between the evolution of the model to be corrected and the evolution of the references. In other words, if the climate change (between calibration and projection) from the model simulations to correct is in disagreement with that from the references, the bias correction method will mostly preserve (i.e., not correct) this "bias of evolution".

For these reasons, in many studies, only the results over the projection period are shown. Here, we also wanted to incorporate results over calibration for comparison.

**Comment 6**

*The evaluation results of the TSMBC using synthetic data based on the VAR process are not convincing (whole section 3) and – at least for me – not fully understandable. For the revisions, I would suggest to leave out this synthetic exercise. Rather, I would focus on better explain the applied methods, i.e. the bias corrections approach applied here (dOTC), the Wetterstein-based metric, and how the naïve RBC (reference approach) works. I am also wondering if this naïve approach is really suitable for fair comparison.*

Response:

The use of synthetic data from a VAR process was mainly (i) to test our TSMBC procedure on understandable data and (ii) to explore the influence of the TSMBC parameters on the results: namely, the choice of a starting row and the effect of the underlying bias correction method within the "row reconstruction method".
A "Vector AutoRegressIve" (VAR) process is a multivariate AutoRegressive (AR) process (i.e., allowing multivariate data) modelling the statistical link between the components of a vector (i.e., multivariate data) when they change in time.
The VAR process is very helpful in this case because the lag is fixed before the experiment, whereas with climate data only an estimation of the maximum lag must be done. So our results about the choice of the starting row and the importance of the dOTC method would not be well argued without this step. The definition of a VAR process is given by Equation (2), and the sampling from this kind of process is performed with this equation: In the case of an order-s VAR process, the first s vectors (i.e., from time 1 to s) are fixed and the VAR process allows generating new values for the components of the vector at time s+1.

The text now reads in section 2.1, before Equation (2):
     *"A VAR process is a multivariate AutoRegressive (AR) process (i.e., allowing multivariate data) modelling the statistical link between the components of a vector (i.e., multivariate data) when they change in time. In the following, a VAR is used to generate multivariate time series…"*

and a few lines after Equation (2):
     *"The sampling is performed based on Equation (2): the first s vectors (i.e., from time 1 to s) are initialised and the VAR process allows generating new values for the d components of the vector at time s+1."*

Regarding the description of the bias correction methods used and the Wasserstein distance, two appendices have been written instead of the previous appendix A: the first one (new appendix A) describes how a bias correction method can be considered as a probability distribution and how dOTC works in this context; the second one (new appendix B) describes the Wasserstein metric.
These appendices are not cited here for sake of space.

Regarding RBC: As explained at the beginning of section 3, the RBC method is necessary to distinguish which part of the correction results comes from the reconstruction by row, and which part from the underlying bias correction method. Moreover, the "Random Bias Correction" (RBC) method has been more detailed at the beginning of section 3:

*"Because the reconstruction step preserves the dependence structure, we propose to test which part of the correction is due to the underlying method (here dOTC), and which part is due to the reconstruction. To do so, a second underlying bias correction method is then used as a benchmark. It corresponds to a very naive method: the correction is randomly drawn from the reference dataset, i.e., for any x ∈ X, the correction is given by a random value y generated according to the distribution of Y. In practice, values from Y are resampled."*

**Comment 7**

*The introduction should be improved, e.g. the statement given in line 28 (… (ii) from inherent biases in the model simulations.") is not very helpful. Potential reasons for the biases shall be mentioned. More and more recent references are required, e.g. for strong statements given in lines 39 & 40.*

Response:

The main reason for biases is the inability of models (whether it is a GCM or a RCM) to exactly reproduce the observations, due to e.g. parameterizations, or processes not or poorly represented.  The sentence regarding "inherent biases" has then be rewritten to mention reasons for the biases as:

"*... (ii) from inherent biases in the model simulations, due to parameterizations, or processes not or poorly represented (e.g., McFarlane, 2011).*"
and the following reference has been added:

"McFarlane, N.: Parameterizations: representing key processes in climate models without resolving them, WIREs Climate Change, 2, 482–497, https://doi.org/https://doi.org/10.1002/wcc.122, https://onlinelibrary.wiley.com/doi/abs/10.1002/wcc.122, 2011."

Regarding the fact that "*the obtained downscaled/bias corrected climate data can then serve as input into impact models*" (lines 39&40 of the initially submitted article), this is indeed a common assumption made in many impact studies. More references have been given to support this statement:

"*The obtained downscaled/bias corrected climate data can then serve as input into impact models (e.g., Teutschbein and Seibert, 2012; Galmarini et al., 2019; Bartók et al., 2019; Chen et al.,2021, among many others*)."

[Figure]

Figure to answer comment 3 from reviewer 2: Maps equivalent to those given in Figure 1 of the article but at the scale of France.

---

## Author Response (AR2)

**"Is time a variable like the others in multivariate statistical downscaling and bias correction?", reviewer 1-2**

Yoann Robin & Mathieu Vrac, https://doi.org/10.5194/esd-2021-12

**Reviewer 1**

**General comments**

The paper presents an original approach to deal with temporal correlation during multivariate bias adjustment of climate variables.

I have 2 main questions:

1. Don't you think that an initial study of the autocorrelation of the variables of interest could guide the choice of the lag?

2. How is the occurrence of rainfall dealt with? Maybe this is the process which bears most of the autocorrelation?

First, we would like to thank this anonymous reviewer for her/his thorough reading and interesting comments. We tried to take them into account and we provide point-by-point responses below in blue.

1. Regarding an initial study of the autocorrelation: In the present article, we have investigated TSMBC for a "fixed lag", but it would be indeed quite relevant to choose the lag according to the specific temporal properties of the variables and the area of interest. As variables like temperature and pressure have a much longer memory than precipitation, the choice of lag should be based on this type of information, as well as on the analysis of the data to be corrected. The following sentence has then been added into the conclusion:

   "*Note also that the chosen lag in TSMBC should be adapted to the type of variable and the area of interest. For example, taking 3 days ($s=3$) for precipitation in Europe seems reasonable, while pressure or temperature could require a week ($s\geq7$). Hence, a preliminary analysis of the autocorrelation or temporal properties of the variables to be corrected should be performed to decide about the relevant lag to use.*"

2. Regarding the occurrence of rainfall: In the TSMBC approach proposed here (i.e., using dOTC as underlying MBC method), the occurrence of rainfall is not treated differently from the non-occurrence. However, it is true that the sequences of dry days and wet days can bear a major part of the autocorrelation information. Hence, it could be interesting to account for this specific aspect of precipitation when performing the underlying MBC method. This is now clarified in the "Conclusion and discussion" section:

*"In addition, when dealing with precipitation, the rainfall occurrence is not treated differently from the non-occurrence (dry days) by the TSMBC approach proposed here (i.e., using dOTC as underlying MBC method). However, the sequences of dry days and wet days can bear a major part of the autocorrelation information. Hence, it could be interesting to account for this specific aspect of precipitation when performing the underlying MBC method."*

Minor comments:

- P8 line 210: time is written twice in "time series"
- P9 line 237: "This generates the corrections"
- P14 line 396: "single" instead of "singles"
- P15 line 427: "has" is written twice in "has been"
- P15 line 430: was then applied or has then been applied

All these technical comments have been corrected. Thank you for pointing them out.

**Reviewer 2**

**General comments**

The manuscript describes an approach to correcting biases in future climate projections based on 'time shifted multivariate bias correction' and the dynamical Optimal Transport Correction (dOTC). I think the overall manuscript is carefully written and the authors examined various aspects of their method to test its robustness and performance.

1. The only issue that I am concerned about is that the current description on 'tas/pr', 'pr/tas', 'pr/pr', and 'tas/tas' pairs are somewhat messed up. These abbreviations for the pairs suddenly show up in Line 330 for the first time in the manuscript without much explanation. The readers may be able to figure out their meanings much later in Lines 425-428, but the difference between 'tas/pr' and 'pr/tas' still remains unclear. How are they exactly different? Does 'tas/pr' mean the lagged cross-correlation between the past temperature and the current precipitation?

2. This suggestion is largely optional: Perhaps a similar approach to the one proposed here can be applicable to a delta change method, which will be particularly useful when high-quality observational data are available and the model fails to capture some important features of the observations. Perhaps authors can comment on this point?

First, we would like to thank this anonymous reviewer for her/his thorough reading and interesting comments. We tried to take them into account and we provide point-by-point responses below in blue.

1. Regarding the "couples" tas/pr, pr/tas, tas/tas and pr/pr: The reviewer is right. The couples tas/pr and pr/tas are respectively the lagged cross-correlations between tas and past pr, and between pr and past tas. We have added the sentence "*The couples tas/tas, pr/pr, tas/pr and pr/tas are, respectively, the correlations between temperature and lagged (in past) temperature, precipitation and lagged precipitation, temperature and lagged precipitation, and precipitation and lagged temperature.*" at the first occurence of the "couples" term (section 4.1, paragraph 4).

2. Regarding the delta change method: This suggestion is not really clear to us. It is not clear at all how a delta change method could be inserted into a TSMBC approach. Indeed, one main issue is that the delta change method is a univariate method while TSMBC requires a multivariate bias correction method. Hence, a delta change does not sound to be appropriate within a TSMBC approach.